# Pathogen-driven degradation of endogenous and therapeutic antibodies during streptococcal infections

Alejandro Gomez Toledo [1,6], Eleni Bratanis [1,6], Erika Velásquez[2], Sounak Chowdhury[1], Berit Olofsson [1], James T. Sorrentino [3], Christofer Karlsson [1], Nathan E. Lewis [4], Jeffrey D. Esko[5], Mattias Collin [1], Oonagh Shannon[1] & Johan Malmström [1] ✉

Group A streptococcus (GAS) is a major bacterial pathogen responsible for both local and systemic infections in humans. The molecular mechanisms that contribute to disease heterogeneity remain poorly understood. Here we show that the transition from a local to a systemic GAS infection is paralleled by pathogen-driven alterations in IgG homeostasis. Using animal models and a combination of sensitive proteomics and glycoproteomics readouts, we documented the progressive accumulation of IgG cleavage products in plasma, due to extensive enzymatic degradation triggered by GAS infection in vivo. The level of IgG degradation was modulated by the route of pathogen inoculation, and mechanistically linked to the combined activities of the bacterial protease IdeS and the endoglycosidase EndoS, upregulated during infection. Importantly, we show that these virulence factors can alter the structure and function of exogenous therapeutic IgG in vivo. These results shed light on the role of bacterial virulence factors in shaping GAS pathogenesis, and potentially blunting the efficacy of antimicrobial therapies.

*Streptococcus pyogenes*, also known as Group A streptococcus (GAS), is a human specific pathogen that causes a broad spectrum of both local and systemic infections[1–3]. The global burden of GAS diseases has been estimated to ~18 million severe cases and ~500,000 deaths each year, constituting a substantial source of morbidity and mortality worldwide[4]. GAS infections display significant heterogeneity regarding tissue tropism, disease severity, and occurrence of post-infectious sequelae. For example, GAS is responsible for a variety of relatively mild and localized skin and throat infections that result in impetigo and pharyngitis. These localized infections are often self-limiting and responsive to antimicrobial treatments, but in some cases, patients may progress into life-threatening invasive conditions, including

sepsis and necrotizing fasciitis[5,6]. GAS infections are also linked to a wide array of severe postinfectious immune-mediated disorders, such as glomerulonephritis and acute rheumatic fever. The basis for this large disease heterogeneity is poorly understood. Defining the underlying factors that modulate bacterial virulence in different disease contexts is instrumental to the development of more sensitive diagnostics and more effective targeted therapeutics.

GAS ability to cause disease is dependent on its capacity to subvert host defenses and to evade immunosurveillance. In particular, GAS has evolved multiple mechanisms to target the structure and function of host immunoglobulin G (IgG), in order to circumvent antibody-mediated responses. For example, GAS surface proteins can

[1]Division of Infection Medicine, Department of Clinical Sciences, Lund University, Lund, Sweden. [2]IPSC Laboratory for CNS Disease Modeling, Department of Experimental Medical Sciences, Lund University, Lund, Sweden. [3]Bioinformatics and Systems Biology Graduate Program, University of California, San Diego, La Jolla, CA, USA. [4]Departments of Pediatrics and Bioengineering, University of California, San Diego, La Jolla, CA, USA. [5]Department of Cellular and Molecular Medicine, University of California, San Diego, La Jolla, CA, USA. [6]These authors contributed equally: Alejandro Gomez Toledo, Eleni Bratanis. ✉e-mail: johan.malmstrom@med.lu.se

sequester antibodies by binding to the IgG Fc-region, preventing bacterial opsonization and phagocytosis via Fcγ-receptor expressing immune cells[7]. This "non-immune" IgG binding by the bacteria is dependent on the local IgG concentration of specific tissue environments, suggesting that the ability of GAS to counteract adaptive immune responses might influence both pathogen tropism and disease phenotypes[8]. Furthermore, GAS expresses at least two proteases, IdeS[9], which specifically cleaves IgGs in the hinge region, and SpeB[10], a broad spectrum protease that degrades Ig antibodies, as well as other protein targets, at multiple sites. Both enzymes reduce the capacity of IgG to elicit downstream Fc-dependent effector functions. In addition, GAS secretes EndoS[11], a bacterial glycan hydrolase that binds to IgG and removes the conserved asparagine N-linked oligosaccharide invariably attached to the Fc heavy chains of both human and murine antibodies. Deglycosylation induces conformational changes that reduce the IgG binding affinity for Fcγ-receptors, which in turn abrogates Fc-mediated protective functions[12]. The fact that GAS has evolved so many different systems to cope with host IgG responses highlights the critical importance of antibodies in mediating protection against streptococcal infections. Moreover, the expression of IgG targeting virulence factors is also a potential concern for the future development of a GAS vaccine, as well as for the evaluation of the efficacy of intravenous Immunoglobulin (IVIG) therapy, a pharmaceutical mixture of polyclonal IgG derived from thousands of individuals, that is advocated as a promising adjuvant therapy for severe streptococcal diseases[13–15].

We have previously reported that GAS infections are associated with IgG glycan degradation in both humans and mice[16]. Interestingly, the degree of deglycosylation was found to be more pronounced at the local site, in patients with superficial skin and throat infections, compared to circulating IgG in the plasma of systemically infected septic patients. These differences suggest that EndoS activity might be modulated by signals derived from the tissue environment and/or associated with progression of invasive disease. However, the exact mechanisms that regulate the activity of EndoS are poorly understood, as well as their contribution to clinically relevant transitions, such as the switch from a local to a life-threatening systemic infection. The interaction of EndoS with other IgG-targeting mechanisms, and their combined impact on therapeutic interventions, such as IVIG therapy, also remain unaddressed. In this study, we provide evidence for the critical role of the host microenvironment in regulating the ability of GAS to induce changes in host IgG homeostasis during an ongoing infection. Homeostatic imbalance is mediated by the simultaneous secretion of IdeS and EndoS into circulation, which in turn results in widespread proteolytic and glycan degradation of both endogenous murine IgG, as well as exogenously administered therapeutic antibodies. More importantly, IgG degradation is modulated by the route of infection, highlighting the critical role of the host microenvironment in shaping both GAS virulence and the therapeutic efficacy of antibody-based antimicrobial treatments.

## Results

### Streptococcal EndoS deglycosylates all IgG subtypes in a murine model of disseminating GAS infection

We have previously reported the accumulation of deglycosylated IgG in the body fluids and tissues of GAS-infected patients, and shown that this can be partially recapitulated in a murine model of disseminating GAS infection[16]. In this previous study, IgG deglycosylation was monitored through targeted selected reaction monitoring (SRM) mass spectrometry (MS), resulting in the absolute quantification of the most abundant truncated glycoform of murine IgG1 at 24 h post infection (p.i.), a single N-acetylglucosamine (GlcNAc) linked to the amide nitrogen of the asparagine residue 297. The specificity of this EndoS-mediated phenotype was confirmed by control infection with isogenic mutant bacteria. However, the high sensitivity but reduced scope of

this approach, limited by the scarce availability of commercial glyco-peptide standards, came at the expense of losing track of potential changes occurring on other IgG isoforms. To generate a more comprehensive view of all ~30 different glycoforms that normally modify the 4–5 different IgG subtypes circulating in mouse plasma, we took advantage of the versatility of label-free glycoproteomics analysis, using data-dependent acquisition (DDA) mass spectrometry and stepped high-energy collisional dissociation (sHCD), which effectively resolves site-specific IgG glycopeptide differences[17]. Here, C57BL/6 J mice were subcutaneously inoculated with $2 \times 10^5$ colony forming units (CFU) of the highly virulent M1 strain AP1, which causes a local skin infection that becomes fully systemic by 24 h, and results in ~50% mortality by 36 h p.i. To assess the extent and time-dependency of the IgG deglycosylation in this model, plasma from infected mice was collected at 0 h, 12 h, 24 h and 36 h after inoculation. Plasma IgG was purified on protein G columns using an automated liquid handling platform, digested into peptides, and subjected to glycoproteomics analysis.

As expected, the data confirmed that GAS infection triggers profound time-dependent IgG glycan remodeling in murine plasma, which results in the generation of truncated glycoprotein products derived from the Fc-region of several murine IgG subclasses and containing a single GlcNAc, with or without a core fucose (Fuc) (Fig. 1a–e). However, infection with an isogenic EndoS mutant GAS strain reduced the truncated glycopeptides to baseline levels confirming that deglycosylation was mainly driven by the EndoS activity (Fig. 1c–f, g). In uninfected conditions, murine IgG was modified with complex-type biantennary N-linked glycans with variable degrees of galactosylation, fucosylation and sialylation, that slightly differed across IgG subclasses, with IgG2b and IgG2c containing more galactose than IgG1 and IgG3 (Fig. 1h–k). Confirming previous reports, murine IgG had trace levels of bisecting GlcNAc[18,19], and terminal sialylation was overwhelmingly accounted for by N-glycolylneuraminic acid (NeuGc) monosaccharides, as opposed to the exclusive expression of N-acetylneuraminic acid (NeuAc) of human glycoproteins, due to the inactivation of cytidine monophospho-N-acetylneuraminic acid hydroxylase (CMAH) during human evolution[20]. In contrast, the predominant IgG glycoforms of infected plasma were truncated glycoprotein products, a cleavage pattern that is consistent with the catalytic mode of action of streptococcal EndoS (Fig. 1h–k). Truncated glycopeptides accounted for a small fraction of the total glycopeptide intensity of each IgG subclass by 12 h (IgG1: 2.5%, IgG2b: 4.0%, IgG2c: 4.4% and IgG3: 11.7%). However, these products significantly accumulated by 24 h (IgG1: 46.5%, IgG2b: 68.4%, IgG2c: 66.0% and IgG3: 88.2%), resulting in an almost complete deglycosylation of the entire murine IgG pool by 36 h p.i. (IgG1: 87.1%, IgG2b: 97.2%, IgG2c: 94.3% and IgG3: 94.2%).

Multiple strategies to directly measure EndoS in infected mouse plasma, including targeted and untargeted mass spectrometry analysis, were unsuccessful, suggesting that the enzyme circulates at low levels. However, the presence of EndoS can be probed by enzymatic assays due to the efficient catalytic activity of the enzyme. To assess the direct involvement of EndoS in the IgG phenotype observed, we performed overnight digestion of uninfected human and murine plasma with recombinant EndoS in vitro. EndoS digestion resulted in a similar deglycosylation pattern as the one observed in vivo, with ~79% of human and ~93% of mouse IgGs being hydrolyzed into EndoS truncated glycoprotein products (Supplementary Fig. 1). The remaining intact IgG glycoforms were mostly assigned to high-mannose N-linked glycans, which are known to be resistant to EndoS activity[21]. Next, we passed infected plasma (36 h p.i.) through Protein G columns and the flow-through containing the non-IgG glycoproteins was collected and subjected to glycoproteomics to determine whether other glycoprotein substrates were also targeted by the deglycosylation activity triggered by GAS. Glycoproteomics analysis

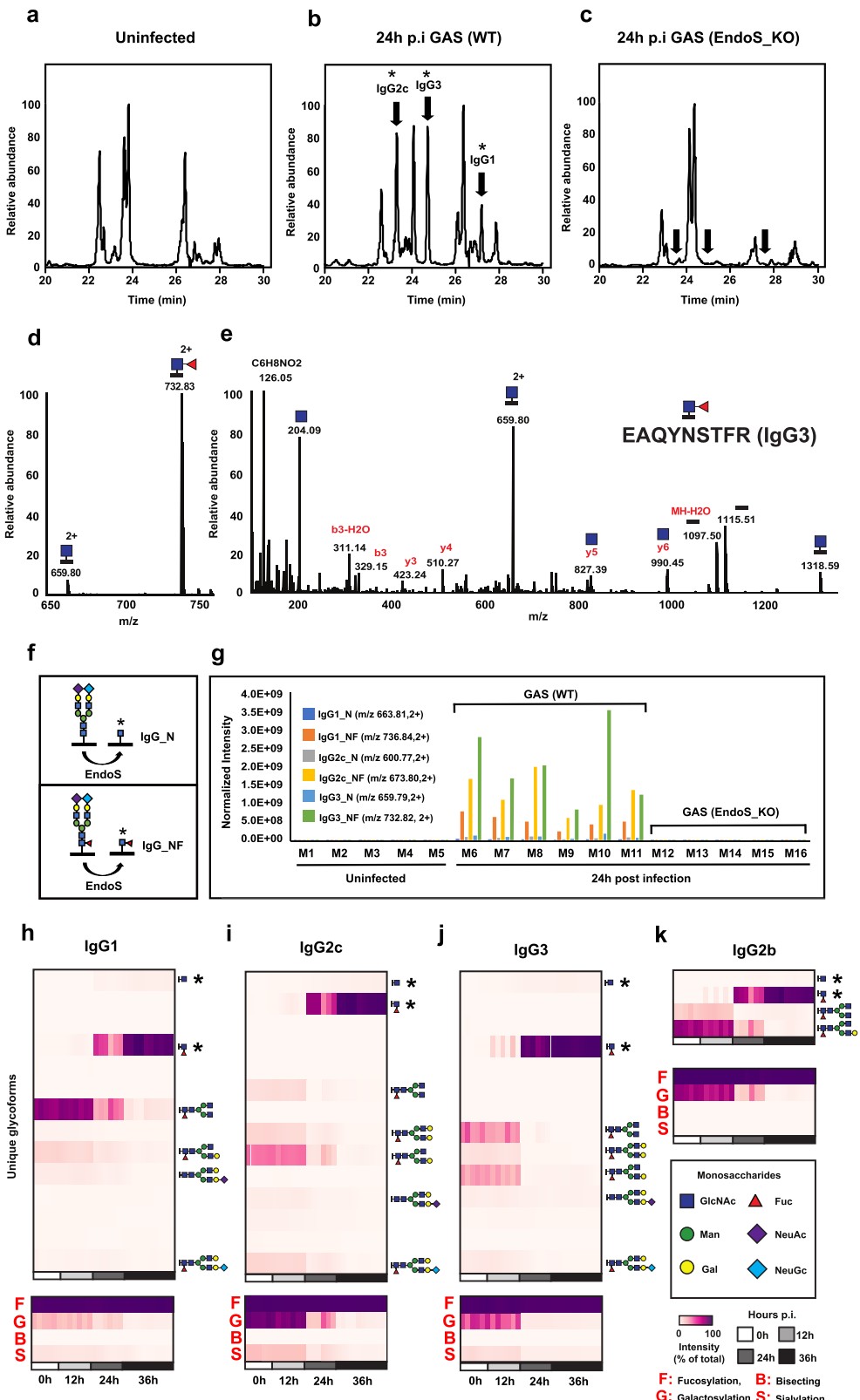

identified 246 glycopeptides derived from 48 abundant plasma glycoproteins with a broad spectrum of complex-type bi- and tri-antennary N-glycan structures. Most glycoproteins were decorated with terminal NeuGc modifications, although a small fraction of NeuAc-containing glycans as well as high-mannose structures were also observed (Supplementary Fig. 2 and Supplementary Table. 1). Importantly, the typical signature of EndoS mediated glycan truncation was only observed on IgG, confirming the high substrate specificity that has been reported and structurally elucidated for EndoS[22]. Taken together, the data is consistent with GAS-infection triggering the time-dependent deglycosylation of all murine IgG subtypes, which results in the plasma accumulation of truncated glycoprotein products with the typical cleavage pattern of streptococcal EndoS activity.

**Fig. 1 | Glycosylation analysis of murine IgGs during subcutaneous GAS infection.** Representative time-resolved elution profiles of murine tryptic IgG glyco-peptides based on extracted ion chromatograms (XIC) of the *N*-acetylglucosamine (GlcNAc) oxonium ion m/z 204.09, generated upon glycopeptide fragmentation in (**a**), uninfected conditions, **b** 24 h post infection with a wildtype AP1 GAS strain, and (**c**), 24 h post infection with an isogenic EndoS mutant AP1 strain. Stars denote differential chromatographic peaks compared to uninfected controls. **d** MS1 pre-cursor intensities of main truncated glycoforms derived from IgG3. **e** Assigned fragmentation pattern of the precursor m/z 732.83 (2 +) corresponding to the murine IgG3 peptide backbone modified with one GlcNAc and one fucose (Fuc). **f** Expected IgG deglycosylation products produced by EndoS enzymatic activity. **g** Quantification of truncated glycopeptide products in wildtype vs mutant bac-terial infections. Relative quantification of the changes in glycosylation patterns of (**h**) IgG1, (**i**) IgG2c, (**j**) IgG3, and (**k**) IgG2b across infected animals ($n$ = 5–10 animals/time point) over the time course of the infection. Truncated glycoforms are high-lighted with stars. The glycan composition of the most abundant glycoforms are drawn. Source data are provided as a Source Data file.

## IdeS activity contributes to alterations in IgG homeostasis

To understand whether glycan degradation is paralleled by other changes in the structure and function of IgGs and/or other murine proteins, the plasma samples taken at different time points (0 h, 12 h, 24 h, and 36 h p.i.) were further subjected to quantitative proteomics analysis. The proteomics data revealed that disease progression was associated with significant changes in the levels of several proteins linked to coagulation and inflammation, and with alterations in the abundance of factors involved in cellular and humoral responses (Supplementary Fig. 3). Principal Component Analysis (PCA) stratified the plasma proteome changes into two distinct stages, one early stage (0 h and 12 h) and one late stage (24 h and 36 h) (Fig. 2a). The early stage was driven by the activation of the acute phase response and other inflammatory proteome changes, whereas the late stage was characterized by the plasma accumulation of several markers of tissue damage, neutrophil proteins, and Proteoglycan 4 (PRG4), a protein that has recently been reported to accumulate both in the plasma and on the injured vasculature of a murine model of *Staphylococcus aureus* (*S. aureus*) sepsis[23,24].

Surprisingly, in addition to the expected decrease in the mole-cular weight of the IgG heavy chains due to deglycosylation (Fig. 2c–g), disease progression in the mouse model was also associated with a significant reduction in the levels of IgG3 (Fig. 2b). Quantification of the individual IgG subtypes in plasma and liver tissues showed that the reduction in IgG concentration was in fact restricted to IgG3 (Fig. 2h), a phenotype that was specifically linked to GAS infection since *S. aureus* bacteremia did not induce changes in IgG levels, neither in the plasma nor in the organs (Fig. 2i). The hinge region of murine IgG3 is report-edly susceptible to proteolytic digestion by streptococcal IdeS[25]. It has also been reported that IdeS degradation of human IgGs is rapidly followed by IgG clearance from circulation[26]. Based on these previous observations, we hypothesized that IdeS may also be responsible for the reduced levels of IgG3 in the mouse model. Overnight digestion of human and mouse plasma with recombinant IdeS revealed that unlike the general proteolytic susceptibility of human IgGs, IgG3 is the only murine IgG expressed by C57BL/6 J mice that can be partially cleaved in the hinge region in vitro (Supplementary Table. 2). This differential susceptibility is linked to distinct amino acid sequence variations in the hinge regions of the IgG subtypes in humans vs mice (Fig. 2j). To determine whether IdeS activity is responsible for the observed reduction of IgG3 in the GAS model, we used mass spectrometry to quantify the levels of the expected IdeS specific proteolytic reporter fragments of the IgG hinge regions in mouse plasma over the course of infection. Intact IgG3 hinge region peptides were readily measured in uninfected as well as infected samples up to 12 h p.i. (Fig. 2k). However, their abundances significantly decreased by 24 h and 36 h p.i., paral-leled by an increase in IdeS-specific cleavage products. These data suggest that IdeS might indeed be responsible for the proteolytic degradation and systemic reduction of IgG3 in this mouse model of disseminating GAS infection. More importantly, they highlight funda-mental differences between mouse and human IgGs in terms of their amino acid sequences and their sensitivity to bacterial proteolytic factors. Collectively, our results demonstrate that disease progression in the GAS model is paralleled by profound alterations in the expression of multiple plasma proteins over time, and a significant IdeS- and EndoS-dependent remodeling of IgG, resulting in proteolytic degradation of IgG3 and complete deglycosylation of the Fc region of all murine IgG subtypes. Finally, although the plasma levels of IdeS and EndoS were below the limit of detection of the mass spectrometry assays, both enzymes were unambiguously identified in murine skin at the site of infection (Supplementary Fig. 4), providing direct evidence for their in vivo expression during the ongoing infection.

## EndoS and IdeS mediate degradation of therapeutic antibodies in vivo

Both EndoS and IdeS are highly conserved across all GAS genomes sequenced to date, indicating that these enzymes might be important for GAS virulence. In previous work, we have demonstrated that EndoS expression interferes with the therapeutic efficacy of active immuni-zation against the protective streptococcal antigen M1[16], but its impact on the efficacy of passive immunization remains unexplored. Given the observed glycan and proteolytic degradation of endogenous IgG in the mouse model, we hypothesized that upregulation of EndoS and IdeS activities might also result in cleavage and inactivation of exogenously administered IgG. To test this hypothesis, we probed the ability of plasma derived from GAS infected mice to degrade IVIG, a pharmaceutical-grade IgG mixture derived from thousands of human donors, and therefore representative of different IgG subclasses and glycosylation patterns. We have previously shown that IVIG contains antibodies that mediate GAS opsonophagocytosis, at least in vitro[27]. Also, based on their distinct amino acid sequences in the Fc-region, murine and human IgG can be separately quantified using mass spec-trometry, facilitating the analysis of complex samples containing IgG from both species. As shown in Fig. 3a–c, EndoS truncated human Fc-glycopeptides were abundantly detected in infected mouse plasma (36 h p.i.) after ex-vivo overnight incubation with human IVIG, indi-cating that the circulating levels of EndoS were sufficient to robustly deglycosylate exogenous human antibodies.

To investigate whether EndoS-driven deglycosylation of IVIG might also occur in vivo, mice were pretreated with IVIG followed by subcutaneous challenge with GAS (Fig. 3d–h). Plasma samples were collected at 36 h p.i., and subjected to IgG enrichment, followed by glycoproteomics analysis. The results showed that EndoS abundantly deglycosylates the administered IVIG in vivo during GAS infection, hydrolyzing ~50–70% of the N-glycans on both human IgG1 and IgG2 (Fig. 3d, e, g). In contrast, infection with an isogenic AP1 EndoS mutant strain[11] prevented the occurrence of EndoS specific truncated glyco-peptides and resulted in maintained intact glycan structures on both human IgG1 and IgG2 (Fig. 3f, g), similar to the glycan structures observed in the uninfected controls. These results demonstrate that EndoS is responsible for the IgG deglycosylation phenotype observed in this model. Notably, despite the marked differences in IgG glycan modifications in the infection with wildtype bacteria compared to isogenic mutant, the total IVIG levels in plasma remained unchanged (Fig. 3h), suggesting that glycosylation does not regulate the half-life of circulating human IgG during infection. Finally, plasma samples were also subjected to mass spectrometric quantification to assess the presence of IgG products consistent with IdeS proteolytic processing

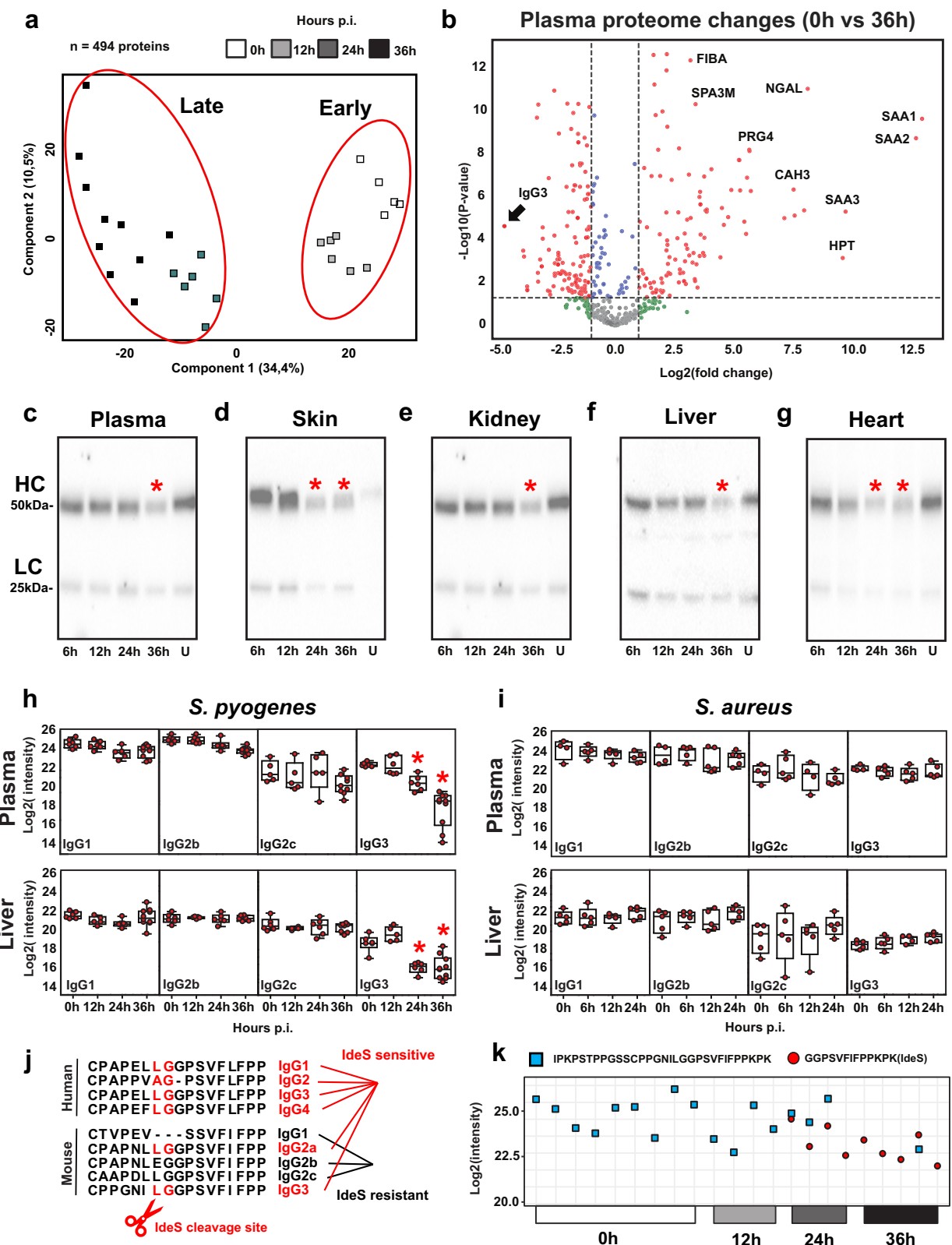

of human IVIG. Accordingly, we identified abundant IdeS cleaved peptides of human IgG in the plasma samples of infected IVIG-pretreated mice, but not in uninfected animals (Supplementary Fig. 5), confirming that both glycosidic and proteolytic mechanisms can simultaneously act on exogenously administered antibodies, leading to substantial remodeling of circulating IVIG.

## GAS virulence in the disseminating infection model is not attenuated by EndoS inactivation or IVIG treatment

Both preclinical and clinical studies have shown variable results regarding the therapeutic benefits of IVIG therapy to treat GAS infections[13,14,28]. In our model of disseminating GAS infection, we found that IVIG treatment did not confer protection against the wildtype AP1 strain or the EndoS isogenic mutant. IVIG pre-treatment had a

**Fig. 2 | IdeS mediates changes in murine IgG homeostasis in a subcutaneous model of local to systemic GAS infection. a** Principal component analysis of murine plasma proteins significantly altered during GAS infection (*n* = 5 mice/ time point). Statistical significance was assessed by analysis of variance (ANOVA) with a permutation-based false discovery rate (FDR) for multiple test correction. **b** Volcano plots displaying the most significantly up- or downregulated plasma proteins at 36 h p.i. compared to uninfected controls. Western blot analysis of IgG levels in (**c**) plasma, (**d**) skin, (**e**) kidney, (**f**) liver and (**g**) heart samples across the course of infection. Representative blots of 2 independent experiments. Mass spectrometric measurements of murine IgG subclasses in plasma and liver homogenates during the course of *S. pyogenes* (*n* = 5 mice/time point) (**h**), and *S. aureus* (*n* = 4–5 mice/time point) (**i**) infections. Red stars denote statistically significant changes. Box boundaries represent first and third quartiles, center line indicates median values. Upper whisker extends from the hinge to the largest value no further than 1.5 * IQR from the hinge (where IQR is the inter-quartile range). The lower whisker extends from the hinge to the smallest value, at most 1.5 * IQR of the hinge. **j** Sequence alignment of the hinge regions sensitive to IdeS cleavage across human and mouse IgG subclasses. **k** Plasma levels of intact and IdeS-cleaved tryptic peptides of the hinge region of murine IgG3 at different time points during the infection. Source data are provided as a Source Data file.

negligible impact on disease progression independently of the bacterial strain used for infection, with both treated and untreated animals displaying significant weight loss from 24 h p.i.(Fig. 4a). Furthermore, the absence of EndoS had no impact on the development of leukopenia or bacterial load in distant organs, independently of IVIG pre-treatment (Fig. 4d–f). Additionally, the plasma proteome changes triggered by the infection with the EndoS mutant were virtually indistinguishable from the changes induced by wildtype bacteria (Fig. 4b, c). These results suggest that despite the significant IgG perturbations induced by GAS infection, genetic inactivation of EndoS or pre-treatment with IVIG is not sufficient to prevent disease progression in this mouse model. In an attempt to explain these results, we investigated whether the used IVIG preparation contained IgG antibodies against GAS antigens, and if antigen-specific IgG could activate cellular signaling through Fcγ-receptors. To this end, we used enzyme-linked immunosorbent assay (ELISA) to show that IVIG contains IgG against several GAS-specific antigens (Fig. 4g–i), and a luciferase reporter cell system, to further demonstrate that these antibodies were capable of robustly activating cellular signaling through Fcγ-receptor IIa (CD32) and IIIa (CD16) in an antigen-specific manner (Fig. 4j, k). As expected, this activity was completely abrogated by pre-incubating IVIG with either recombinant IdeS or EndoS, highlighting the ability of both enzymes to block Fcγ-dependent functions, such as antibody-dependent phagocytosis (ADP) and antibody dependent cellular cytotoxicity (ADCC). Based on these results, we concluded that IVIG, despite containing functional GAS-specific IgGs with the capacity to trigger downstream effector functions, did not result in therapeutic benefit, at least in this infection model.

**The route of bacterial inoculation impacts both GAS infection and IVIG therapeutic efficacy**

Despite this apparent lack of therapeutic benefit of IVIG in our infection model, we noticed that both the severity of the infection, as well as the accumulation of IdeS and EndoS cleavage products, became more pronounced towards 24 h p.i., concomitantly with the transition from a contained local, to a systemic infection. We hypothesized that the host tissue environments (skin vs blood) might have an impact on GAS infection, for example by regulating bacterial virulence and susceptibility to treatment. To test this notion, mice were inoculated with the AP1 strain intraperitoneally (IP), to bypass the local skin stage of the infection, and to recreate a single stage systemic infection, as opposed to a two-stage local to systemic infection model. IVIG pre-treatment was also added to one group to evaluate the impact of the route of infection on the therapeutic efficacy of IVIG. Blood and tissues were harvested at 24 h p.i., and subjected to mass spectrometry-based proteomics and glycoproteomics analysis. These analyses revealed the absence of an imbalance in IgG homeostasis in the single stage systemic IP model compared to the two-stage subcutaneous model, with no observable differences in Fc-glycan patterns (Fig. 5a–c) or IgG3 levels (Fig. 6g) between uninfected and infected animals. Additionally, there was no difference in the glycan pattern of the injected human IVIG (Fig. 5d). To rule out endogenous inhibitory factors in plasma that might interfere with the activity of the bacterial enzymes, we look for the presence of human antibodies against EndoS and IdeS in the IVIG formulation. Although both anti-EndoS and anti-IdeS antibodies were detected in IVIG, their partial degradation in the subcutaneous model already indicated that they were of insufficient titers or activity to neutralize the enzymes (Supplementary Fig. 6). Moreover, the total absence of neutralizing murine antibodies triggered by the infection was also ruled out by ELISA measurements, which is also consistent with the rapid disease progression and the lack of adaptive immune responses in both models. Additionally, spiking in recombinant EndoS into IP infected blood resulted in abundant glycan degradation, ruling out the presence of other endogenous inhibitors (Supplementary Fig. 7). On the other hand, this lack of EndoS and IdeS enzymatic activity was consistent with a reduction in the mRNA expression of both EndoS and IdeS in the infected splenic tissues of the IP model compared to the subcutaneous model, pointing towards reduced enzyme expression linked to the route of infection as the most likely explanation for the absence of enzymatic activity in the IP model (Fig. 5e).

In sharp contrast to the results obtained in the two-stage subcutaneous model, pre-treatment of the single stage IP model with IVIG substantially reduced the levels of splenic colonization, with a similar trend also observed for other organs such as the liver, but without having a significant effect on weight loss (Fig. 6a–c). Further therapeutic benefit in the single stage IP model was evidenced by a significant reduction in leukocytopenia by 24 h p.i. (Fig. 6d). Plasma proteome analysis of infected animals revealed that multiple proteins were differentially regulated by the IVIG treatment compared to the untreated infected group (Fig. 6e). These proteome changes were linked to an increase in the plasma levels of liver-derived intracellular metabolic proteins that can leak out to plasma due to inflammation and tissue damage (Fig. 6f). This effect on the plasma proteins was selective since IVIG treatment did not affect the induction of acute phase proteins or vascular activation (Fig. 6h), but significantly reduced the levels of circulating markers of liver damage (Fig. 6i), pointing towards a potential hepatoprotective role for IVIG in this model. In conclusion, our results suggest that the transcriptional status of the bacteria is impacted by the route of infection, which directly implicates the tissue microenvironment in the modulation of GAS pathogenesis through the regulation of the expression of specific virulence factors such as IdeS and EndoS in vivo (Fig. 7). More importantly, the data also highlights how the progression of GAS infection, as well as the therapeutic efficacy of antibody-based treatments, are both fundamentally shaped by the complex interplay between host and pathogen factors.

## Discussion

The specific roles of the multiple IgG-targeting factors expressed by GAS, and their important contributions to bacterial virulence, are only partially understood. This study sheds light on these contributions by showing that both human and murine IgG can be proteolytically digested and deglycosylated by streptococcal IdeS and EndoS secreted during an ongoing infection. Collectively, our observations provide support for the hypothesis that the clinical heterogeneity of GAS

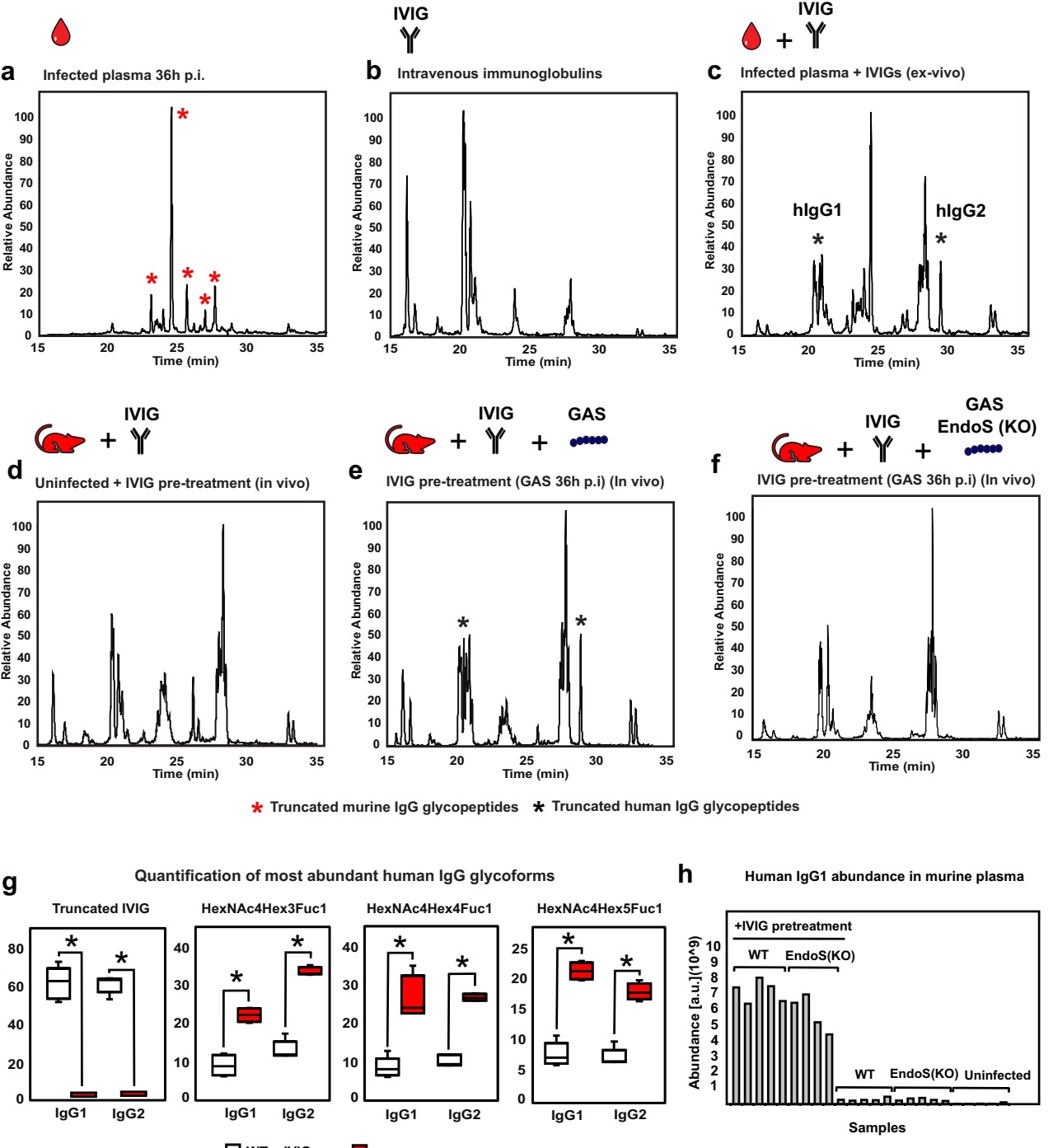

**Fig. 3 | EndoS drives deglycosylation of human IVIG.** Representative elution profiles of IgG glycopeptides based on extracted ion chromatograms (XIC) of the *N*-acetylglucosamine (GlcNAc) oxonium ion m/z 204.09, and purified from (**a**), infected mouse plasma 36 h p.i., (**b**) pharmaceutical-grade IVIGs, (**c**) infected mouse plasma spiked with IVIG and incubated ex-vivo, (**d**) plasma from IVIG-treated mice, (**e**) IVIG-treated mice challenged with wildtype GAS, (**f**) IVIG-treated mice challenged with EndoS (KO) GAS. Red stars denote truncated mouse IgG; black stars denote truncated human IgG. **g** Quantification of human IgG1 and IgG2 Fc-glycopeptides in plasma at 24 h p.i. and derived from mice infected with wildtype vs EndoS (KO) GAS strains and treated with IVIG ($n = 4$ mice/condition). Statistical significance is denoted with starts and was assessed by two-sided Student's t-test ($p < 0.05$). Upper whisker extends from the hinge to the largest value no further than 1.5 * IQR from the hinge (where IQR is the inter-quartile range). The lower whisker extends from the hinge to the smallest value, at most 1.5 * IQR of the hinge. Data beyond the end of the whiskers are called outliers and are plotted individually. **h** Human IgG levels circulating in mouse plasma.

diseases, as well as the efficacy of antibody-based therapies, might both be affected by bacterial expression of IgG-targeting virulence factors.

One of the key findings of this study is that GAS virulence is partially modulated by the host microenvironment. Using a murine model of disseminating GAS infection, we demonstrate that the transition from a local to a systemic infection is here associated with changes in IgG homeostasis, linked to bacterial expression and secretion of IdeS and EndoS into circulation. Surprisingly, bypassing the local stage of the infection significantly reduced the level of systemic IgG

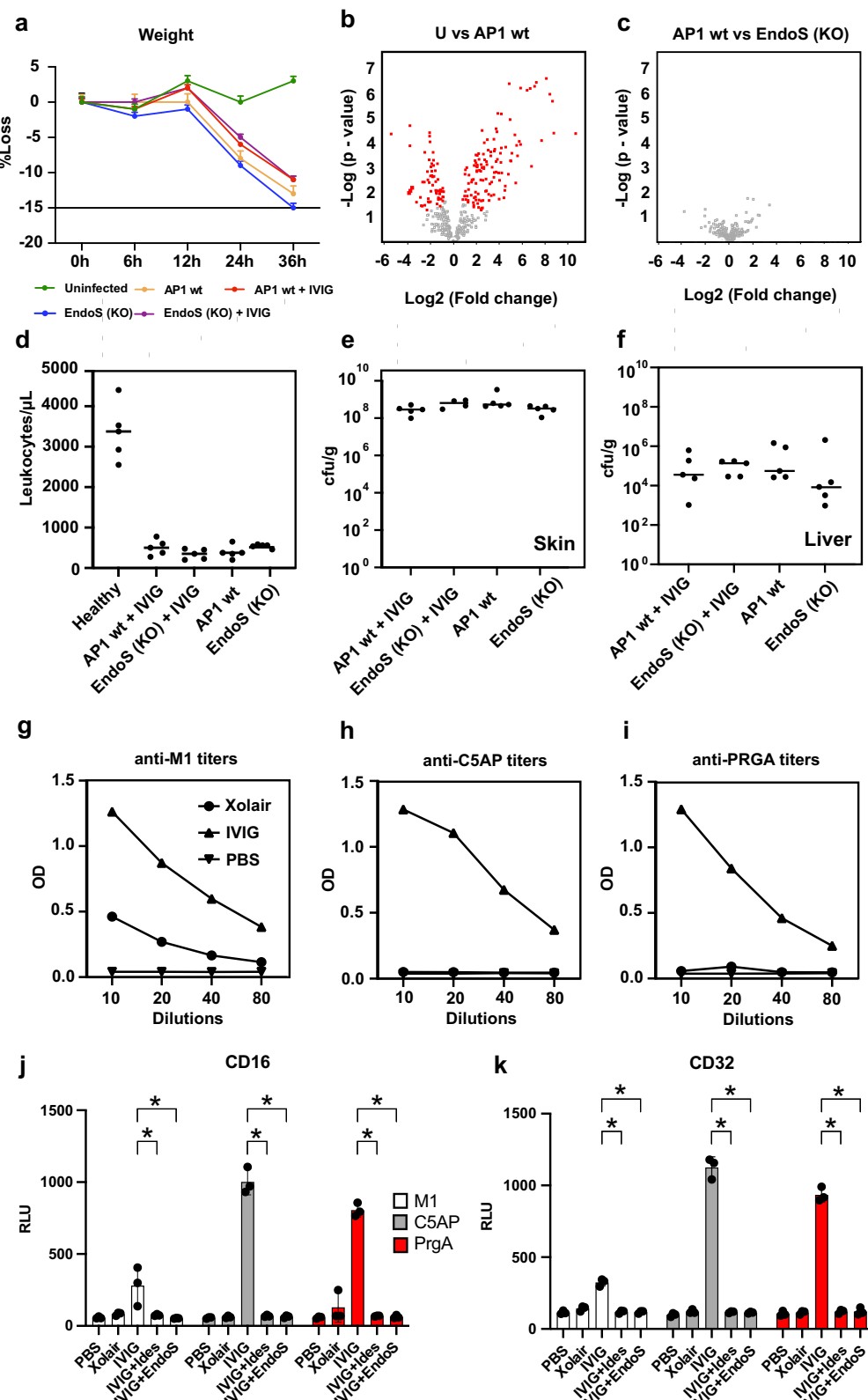

**Fig. 4 | GAS virulence in the subcutaneous infection model is not impacted by IVIG treatments or EndoS-inactivation. a** Time-dependent weight measurements of infected mice during the course of infection ($n = 5$ animals/ time point), Data are presented as mean values +/- SEM. **b** Volcano plots of changes in plasma proteins between uninfected and infected mice with wildtype bacteria, and (**c**) between animals infected with wildtype vs EndoS (KO) GAS strain at 36 h p.i. where red spheres indicate significantly changed protein abundances. **d** The levels of plasma

leukocyte counts, bacterial burden in (**e**), skin, and **f**, liver across the experiments. IVIG antibody titers against GAS-specific (**g**) M1, (**h**), C5A peptidase and (**i**) surface exclusion protein PRGA, as determined by ELISA. FCγ-receptor activity assay of (**j**), CD16 (FCγ IIIa) and (**k**), CD32 (FCγ IIa). Both ELISA and cell assays were done in triplicates in two independent experiments, and statistical significance is denoted with stars and was assessed by two-sided Student's t-test, $p < 0.05$. Data are presented as mean values +/- SEM. Source data are provided as a Source Data file.

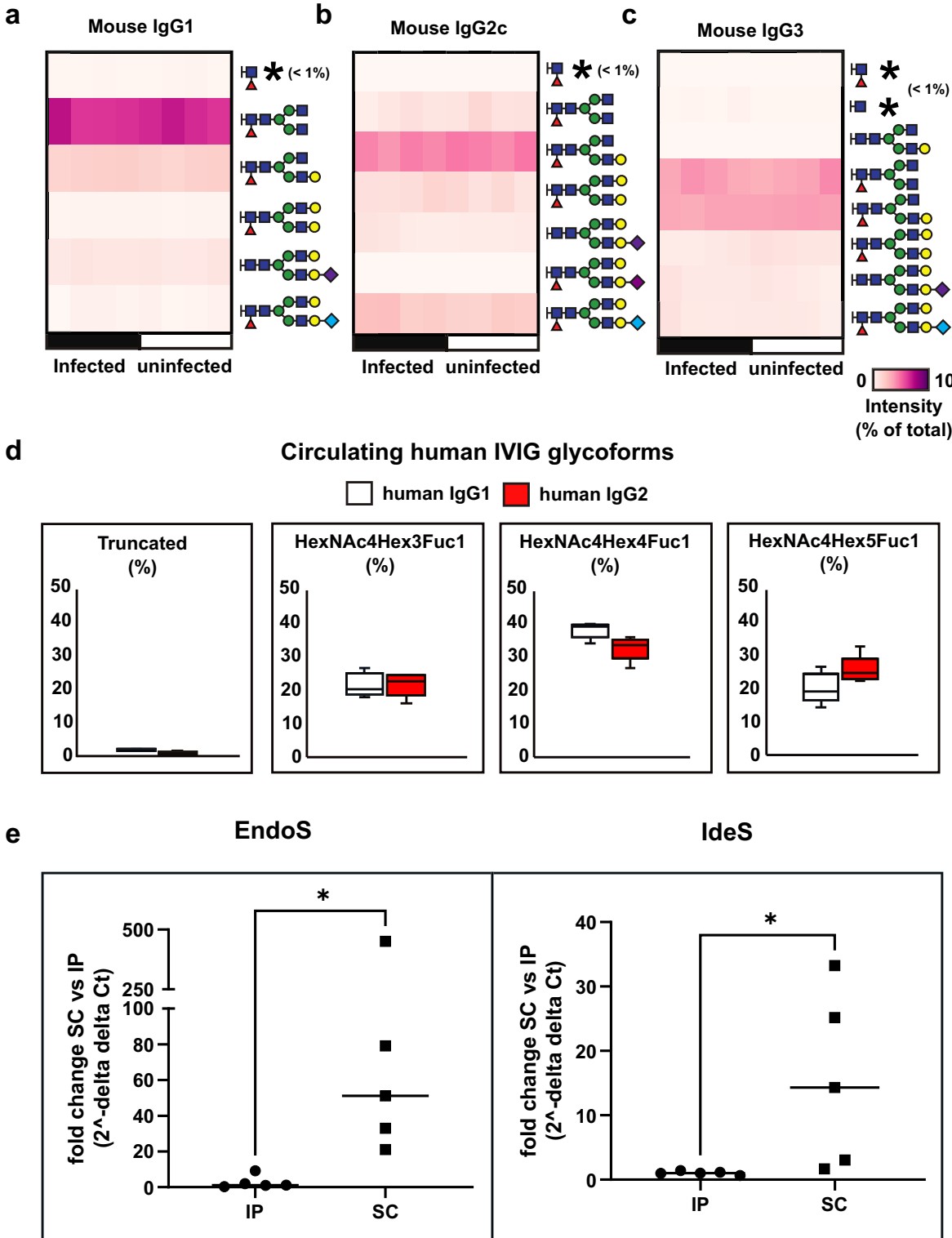

**Fig. 5 | EndoS is not catalytically active in the murine model of intraperitoneal (IP) GAS infection.** Animals were infected by GAS strain AP1 with or without IVIG treatment and blood plasma was harvested after 24 h p.i. Glycopeptide analysis of murine plasma glycoforms for **a**, IgG1. **b** IgG2c and (**c**) IgG3 before and after 24 h IP-infection. **d** Mass spectrometry-based glycopeptide quantification of human IVIG circulating in mouse plasma at 24 h p.i. (*n* = 5 mice/condition). Upper whisker extends from the hinge to the largest value no further than 1.5 * IQR from the hinge (where IQR is the inter-quartile range). The lower whisker extends from the hinge to the smallest value, at most 1.5 * IQR of the hinge. **e** qPCR analysis of the expression of EndoS and IdeS in microbial mRNA isolated from spleens of intraperitoneally or subcutaneously infected mice at 24 h postinfection (*n* = 5/condition). Statistical significance was assessed by two-tailed Mann-Whitney test, *\*p* < 0.05.

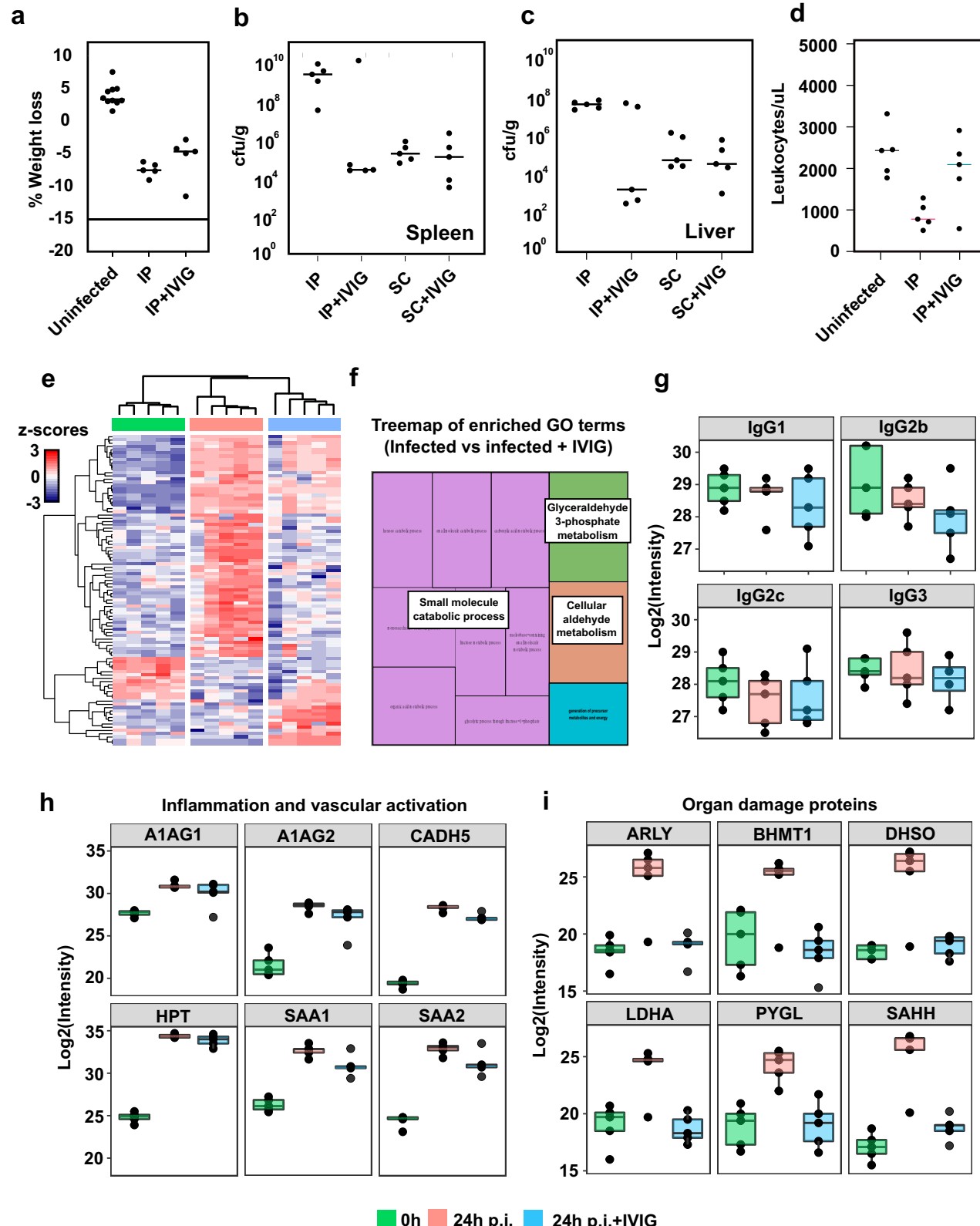

degradation, singling out the tissue microenvironment as an important modulator of these phenotypic changes. Intensive research during the last years has revealed multiple mechanisms that might underpin the ability of the bacteria to switch from triggering local skin and naso-pharyngeal infections to causing life-threatening invasive disease. Among these, the differential expression of virulence factors, alone or in conjunction with acquired mutations in two-component signal

transduction systems, such as *covR/S*, have emerged as important drivers of enhanced bacterial virulence[29–31]. The AP1-strain used in this study is a CovS-mutant that makes the whole infection more potent in the mouse, as opposed to the overall reduction in virulence observed in most GAS strains, since the mouse is not an optimal host for GAS. Although there is a possibility that new mutations might be acquired during our infections, this is most likely to occur over several passages

**Fig. 6 | Murine GAS infection and IVIG therapeutic efficacy is modulated by route of infection. a** Weight measurements of uninfected (*n* = 10) and intraperitoneal (IP) infected mice at 24 h p.i. with or without IVIG pretreatment (6 h before infection) (*n* = 5 mice/condition). **b** Splenic and (**c**) hepatic bacterial burden in IP vs subcutaneous (SC) infection models and IVIG treatment effects. **d** Stabilizing effect of IVIG in leukocyte counts. **e** Hierarchical clustering of plasma proteins significantly altered by IVIGs in IP infected mice. Colors on the top indicate treatment group as shown in the legend at the bottom. **f** TreeMap representation of functional enrichment analysis of plasma proteins differentially regulated by IVIG treatment. Size of the squares is proportional to degree of enrichment for the highlighted

pathways. **g** Mass spectrometric quantification of murine IgG subclasses in the plasma samples. **h** Quantitative analysis of representative plasma proteins involved in inflammation and vascular activation. **i** Quantitative analysis of representative metabolic proteins leaking out to plasma due to organ damage. (*n* = 5 mice/condition). Upper whisker extends from the hinge to the largest value no further than 1.5 * IQR from the hinge (where IQR is the inter-quartile range). The lower whisker extends from the hinge to the smallest value, at most 1.5 * IQR of the hinge. Statistical significance was assessed by analysis of variance (ANOVA) with a permutation-based false discovery rate (FDR) for multiple test correction. Source data are provided as a Source Data file.

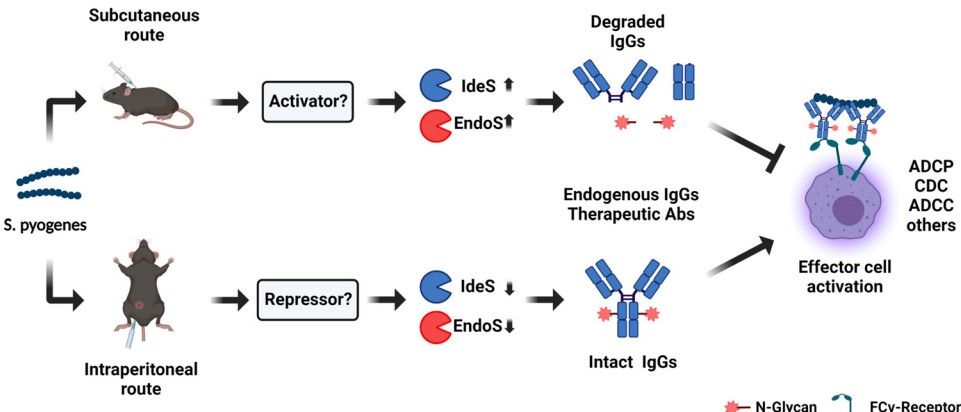

**Fig. 7 | Schematic summary of the mechanisms potentially linking transcriptional changes in GAS to the route of infection, resulting in the differential** secretion of EndoS and IdeS into circulation and their subsequent effect on endogenous and therapeutic IgG antibodies. Created with BioRender.com.

of the same strain on the mice, which was not the case for our particular experimental design. In addition, a tight connection between nutrient sensing and the activation of transcriptional signatures linked to virulence has also been demonstrated. One notable example is the ability of transcriptional regulators such as the Carbon Catabolite Protein A (CCPA), to mediate bacterial adaptation to low glucose environments, such as human skin and tonsils, followed by the upregulation of multiple well-established virulence factors[32,33]. Intriguingly, both IdeS and EndoS are responsive to CCPA signaling, and their expression is also increased in various *covR/S* mutants[32]. Whether these mechanisms also contribute to the phenotypic differences between the subcutaneous and the IP-model remains to be determined, but the differential nutrient availability such as low glucose in the skin vs higher glucose levels in plasma, and their known impact on transcriptional regulators such as CCPA or covR/S could potentially explain these differences. Finally, we cannot rule out that other host factors might also contribute to the modulation of the observed phenotypic switches, including the presence of skin-specific factors (e.g., specific lipids or carbohydrates), or the differential upregulation of host proteases derived from neutrophils or other immune cell populations. However, our conclusion that the IgG degradation is primarily driven by EndoS and IdeS is supported by our extensive in-vivo and in-vitro data, that is also consistent with the exquisite specificity of both enzymes, which has been recently characterized in detail by crystallographic studies[34]. More importantly, phenotypic differences linked to the route of infection and the host microenvironments have been documented both in GAS and other pathogens, indicating that context-sensitive regulation of the bacterial proteome in vivo might be a general phenomenon that contributes to the observed disease heterogeneity of bacterial infections[35–37].

Our findings further indicate that a more careful preclinical evaluation of the integrity of therapeutic antibodies, elicited through active or passive immunization against GAS, is also needed. Inasmuch

as the expression of pathogen-derived IgG-targeting factors might differ across animal models of infection, the efficacy of antibody-based treatment might also fluctuate. These concerns could in principle be extended to preclinical studies of other bacterial infections. Indeed, human pathogens such as *Enterococcus faecalis*[38] and *Streptococcus dysgalactiae*[39] can also express endoglycosidases that target IgG, as well as other host glycoproteins, although their contributions to pathogen virulence during infection remain poorly understood. Also, serotype M49 GAS strains are known to secrete an EndoS-like protein known as Endos2 (instead of EndoS), displaying endo-β-N-acetylglucosaminidase activity on all N-linked glycans of IgG, as well as on biantennary and sialylated glycans of α1-acid glycoprotein[40]. The AP1 strain used in this study does not express EndoS2, but it would be interesting to assess the activity and in vivo regulation of EndoS2 during infections, and how it compares to EndoS regarding substrate specificity, catalytic efficiency, sensitivity to regulatory cues from the microenvironment, and interaction with antibody-based antimicrobial therapies.

In this study, we used pharmaceutical-grade IVIG as a molecular probe to quantify the extent of IgG degradation triggered by GAS during infections. However, the clinical efficacy of IVIG as an adjuvant therapy for systemic bacterial infections and sepsis, has also been the subject of intense investigations. So far, clinical trials have failed to demonstrate a clear benefit of IVIG treatment in decreasing sepsis mortality, except in the specific case of invasive GAS disease and the streptococcal toxic shock syndrome (STSS)[41]. The protective mechanisms of IVIG in the context of GAS diseases are most likely complex, and include both neutralizing toxins and superantigens, as well as increasing bacterial clearance through GAS-specific opsono-phagocytic antibodies[14]. Indeed, here we show that commercial IVIG contains GAS-specific antibodies with the capacity of eliciting efficient Fcγ-signaling through various receptors. However, our data also shows that these activities are completely abrogated by either IdeS or EndoS

treatments, highlighting the potential impact of these pathogenic mechanisms of immune evasion on GAS pathogenesis. The activities of these bacterial enzymes also underline the importance of the IgG protein structure in general, and glycosylation in particular, in mediating some of the Fc-mediated downstream effector functions of IVIG.

Notably, the rapid disease progression of our mouse models (both subcutaneous and intraperitoneal infections) results in mortality by ~24–36 h post infection, and does not allow enough time for the development of an effective adaptive immune response. On the other hand, humans might indeed develop antibodies against GAS through natural exposure, but whether such antibodies can neutralize these enzymes is presently unclear. In fact, previous studies have shown that most people make IgG against IdeS and EndoS, but these antibodies do not interfere with the activity of the enzymes[42]. Similarly, our present data also shows that IVIGs are readily degraded by IdeS and EndoS, suggesting that these IgGs are incapable of blocking and neutralizing the enzymes. Possible reasons for this incapacity might be the occurrence of generally low antigen-specific IgG titers, suboptimal binding and/or reduced capacity of triggering downstream effector functions. Glycan engineering has shown promising results in increasing the anti-inflammatory properties of IVIG-formulations, which has been successfully tested to treat various animal models of inflammatory diseases[43]. The use of similar approaches to boost the efficacy of IVIG in the context of GAS and other bacterial infections remains unexplored, but could in principle lead to similar therapeutic benefits. Also relevant to the antimicrobial effects of IVIG, we found that its therapeutic benefits are modulated by the route of infection. Similar to a previous study using a model of streptococcal toxic shock syndrome[14], we found that IVIG was effective at lowering bacterial spreading and reducing systemic inflammation in a single stage systemic IP model of sepsis, but not if the pathogen was firstly administered subcutaneously to establish a local skin infection, followed by systemic dissemination. The molecular basis for these notable differences is unknown, but could be related to differential contribution of immune cells in skin vs plasma that facilitate clearance. Nevertheless, it is interesting that the therapeutic benefits of IVIG were also linked to specific changes in plasma proteins associated with organ dysfunction, which might indicate an organoprotective function for IVIG in specific disease contexts, perhaps related to a lowering of the dissemination of bacteria into the organs.

Finally, GAS is a human pathogen that suffers from reduced virulence in the mouse, and key factors involved in human infections do not always contribute to disease in the murine setting. For example, secreted GAS superantigens are potent bacterial mitogens that overstimulate T-cell lymphocytes to massively release pro-inflammatory cytokines in humans but not in mice, due to their specificity for the human Major Histocompatibility Complex II (MHC-II)[44]. Similarly, GAS streptokinase (SK) is a virulence factor that activates host plasminogen to facilitate pathogen spreading and dissemination from the initial site of infection into the surrounding tissues. SK significantly contribute to GAS infection in humans, but its virulence is generally diminished in mouse models due to a reduced specificity for murine compared to human plasminogen[45]. In the same line, our current study highlights important species differences between human and mouse immune responses, including the expression of different IgG subclass distributions (i.e., IgG1-4 in humans vs IgG-1, -2b, -2c and -3 in mice) and Fc-glycosylation patterns (e.g., absence of NeuGc in humans), as well as the differential susceptibility of specific IgG subtypes to streptococcal virulence factors such as IdeS. In fact, unlike human IgGs, most murine IgGs are known to be resistant to the proteolytic activity of IdeS, with the notable exception of IgG3[46], which we also found to be cleaved in the subcutaneous model of disseminating GAS infection, and subsequently cleared out from circulation. The basis for the species preferences of IdeS might lie in acquired amino acid variations across the hinge regions of human vs mouse IgG, but more studies are needed to

clarify this, as well as the potential contribution of specific substrate residues to the catalytic efficiency of IdeS. The selectivity of certain GAS virulence factors for host-specific determinants most likely reflects a long history of antagonistic host-pathogen coevolution, given that humans are the only known natural reservoirs of GAS, and the host immune system is the most important source of selective pressure driving GAS evolution[47]. However, in sharp contrast to IdeS, we found that EndoS was able to deglycosylate all human and murine IgG subclasses equally well. These contrasting behaviors suggest that the evolutionary mechanisms shaping the specificity of IdeS and EndoS might be different, despite both enzymes sharing a common substrate. Our understanding of the natural history of GAS has significantly increased during the last years, but future studies are still required to fully understand the evolution of pathogen virulence, and to unravel the molecular mechanisms linking the evolution of host-specific immune responses to the evolution of pathogen-driven immune evading mechanisms.

## Methods

### Ethics statement
All animal use and procedures were approved by the local Malmö/Lund Institutional Animal Care and Use Committee, ethical permit number 03681-2019.

### Bacteria and culture conditions
*Streptococcus pyogenes* AP1 (from the Collection of the World Health Organization Collaborating Center for Reference and Research on Streptococci, Prague, Czech Republic) was grown in Todd–Hewitt broth, supplemented with 0.2% yeast extract (THY, BD diagnostic), overnight (o/n) at 37 °C and 5% $CO_2$, and the isogenic *ndoS* mutant MC14[11] was grown under the same conditions with the addition of 150 µg/ml of kanamycin. Methicillin-resistant *Staphylococcus aureus* (MRSA USA300 TCH1516) was grown as previously reported[23].

### Bacterial infections
*S. pyogenes* AP1 was grown to logarithmic phase in THY medium (37 °C, 5% $CO_2$). Bacteria were washed and resuspended in sterile PBS. Nine-week-old female C57BL/6 J mice (Janvier, Le Genest-Saint-Isle, France) were infected with 50 µl ($2 \times 10^5$ cfu) bacteria by subcutaneous injection on the right flank, as previously reported, or 100 µl ($1 \times 10^7$ cfu) by intraperitoneal injection. Control groups were similarly injected with sterile saline ($n = 9$). Mice were rehydrated subcutaneously with saline 24 h post infection. Bodyweight and general symptoms of infection were monitored regularly. Any animals exhibiting >15% weight loss from pre-infection weight were sacrificed as non-survivors. Otherwise at predetermined time points mice were sacrificed and multiple organs were collected. *S. aureus* intravenous infection was conducted as previously reported ($n = 5$/time point). Blood was taken by cardiac puncture, and collected in tubes containing sodium citrate (Mini-Collect tube, Greiner Bio-One).

### Intravenous immunoglobulin (IVIG) treatments
IVIG (Octagam, 1 mg/ g of body weight) was given by intraperitoneal administration 6 h prior to subcutaneous ($n = 10$) or intraperitoneal ($n = 10$) infection. Control groups were similarly injected with sterile saline ($n = 10$). Bodyweight and general symptoms of infection were monitored regularly. Mice were sacrificed 24 h post infection, and multiple organs were harvested. Blood was taken by cardiac puncture, and collected in tubes containing sodium citrate (MiniCollect tube, Greiner Bio-One).

### Plasma and organ preparations
Citrated blood collected from infected and control mice was centrifuged (2000 x g, 10 min) to obtain platelet free plasma. Plasma was aliquoted and stored at −80 °C. Collected organs were homogenized

(MagnaLyzer, Roche) in PBS using sterile silica beads (1 mm diameter, Techtum). Skin samples were obtained by punch biopsy (10 mm, Acu punch). Degree of bacterial dissemination was determined by serial diluting and plating organ homogenates onto blood agar plates. Colony forming units were counted following o/n incubation (37 °C, 5% CO$_2$), and are presented as cfu/ g of tissue. Remaining organ homogenates were centrifuged (20,000 x g, 10 min, 4 °C) and supernatants immediately transferred and aliquoted into fresh vials. All samples were stored at −80 °C until further analysis. Protein concentration was determined using standard BCA assay (Thermo Scientific) according to manufacturer instructions.

## Flow cytometry of blood cells

Citrated blood collected from infected and control mice was diluted with HEPES buffer containing Mouse BD Fc-block (BD Pharmingen). Total leucocytes were gated according to characteristic forward and side scatter and platelets were identified using anti-CD41 FITC (BD Pharmingen). Antibodies were diluted 1:200 and incubated (15 min, RT). Samples were lysed using 1-step Fix/Lyse Solution (e-Bioscience), washed (500 x g, 5 min) cellular pellets were resuspended in PBS. The samples were analyzed using an Accuri Plus C6 Flow Cytometer (BD Biosciences), and the data was analyzed using C6 Software (BD Biosciences).

## IgG western blots

Samples were normalized by protein concentration, determined using standard BCA assay, and separated on SDS-PAGE (Mini-protean TGX stain-free gels, 4–15% acrylamide, BioRad) under reducing conditions. Proteins were transferred to PVDF membranes using the Trans-Blot Turbo kit (Bio-Rad) according to the manufacturer's instructions. Membranes were blocked with 5% (wt/vol) skimmed milk in PBST (PBS, 0,1% Tween 20) followed by incubation with rabbit anti-mouse IgG (1:2500, Bio-rad). The membranes were washed, followed by incubation with a secondary antibody (goat anti-rabbit HRP-conjugated antibody; Bio-Rad). The membranes were developed using Clarity Western ECL substrate (Bio-Rad) and visualized with the ChemiDoc MP Imager.

## Enzyme-linked immunosorbent (ELISA) assay

To measure GAS specific antibodies in IVIG, 96 well Nunc microtiter plates were coated with 100 μl of recombinant M1, C5AP and PrgA (5 μg/ml) overnight at 4 °C followed by PBST (PBS + 0.05% Tween 20) wash. For blocking, 2% bovine serum albumin (BSA) (100 μl/well) in PBST was added for 30 min at 37 °C and then washed with PBST. The plates were then incubated at 37 °C for 1 h with Xolair (1:10), IVIG (1:10) and PBS (1:10) in dilution series in triplicates, followed by PBST wash. 100 μl/well of protein G-HRP (1:3000, Bio-Rad) was incubated for 1 h at 37 °C, washed with PBST, after which the reaction was developed with 100 μl/well ABTS (20 ml Na-citrate pH 4.5 + 1 ml ABTS + 0.4 ml H202) for 30 min and the OD was measured at 550 nm.

## IgG digestion with recombinant enzymes or septic plasma

Mouse and human plasma were diluted with PBS and incubated with recombinant EndoS or IdeS (enzyme: substrate ratio 1:10) overnight at 37 °C. EndoS cleaved plasma was further subjected to IgG pulldowns as described below, whereas the IdeS treated samples were directly subjected to trypsin digestion. For probing the capacity of septic plasma to digest IgG, roughly 100 μg IVIG (Octagam) was incubated overnight at 37 °C with 10 μl septic plasma (36 h p.i.) and diluted with PBS to a final volume of 100 μl. The cleaved samples were also subjected to IgG pulldowns.

## FCγ- luciferase reporter cell assay

To investigate the role of EndoS and IdeS treated antibodies in triggering antibody-dependent cellular cytotoxicity (ADCC) and antibody-dependent cell-mediated phagocytosis (ADCP) Jurkat-Lucia NFAT-CD16 and CD32 cells (InvivoGen) were used. Nunclon delta surface plates (Thermo Scientific) were coated with 100 μl of 5 μg/ml of M1, C5AP and PrgA overnight at 4 °C. After PBS wash 100 μl of different antibody sources (100 μg/100 μl) were added and incubated for 1 h at 37 °C, followed by PBS wash. 200 μl of CD16 and CD32 cells (100,000 cells/100 μl) in IMDM with 10% heat-inactivated fetal bovine serum (FBS) and Pen-Strep (100 U/ml-100 μg/ml) were added for 6 h at 37 °C. After centrifugation at 150 g for 10 min, 20 μl of the supernatant was added to 50 μl of QUANTI-Luc (InvivoGen) in opaque microtiter plates and the luciferase activity was measured in luminometer.

## Microbial mRNA extraction and qPCR analysis

Infected mouse spleens were harvested and stored in stored at −80 °C in RNAlater (ThermoFisher). The RNAlater was removed and replaced by 1x PBS. The spleens were homogenized using a bead bug machine (6000 x g, 30 s, 2 cycles). The homogenates were treated with 1 μL of PlyC (20 U/μL, generous gift of Prof. Mattias Collin, Lund University) for 30 min at 37 °C to break down the Streptococci. The total RNA of both mouse spleen cells and bacteria was purified by phenol/chloroform extraction. Briefly, 0.5 mL of organ/bacteria homogenate was mixed with 0.8 mL of TRIzol (ThermoFisher) and sheared twice through a 26 gauge needle. Then, 0.18 mL of chloroform was added, vortexed and incubated at RT for 3 min. The samples were centrifuged at 12,000 x g, 15 min, 4 °C resulting in a phase separation. The aqueous phase was transferred to a fresh tube, 0.4 mL isopropanol added, left to incubate for 10 min at 10 min, followed by another centrifugation at 12,000 x g, 10 min, 4 °C. The RNA pellet was washed twice in 75% ethanol and finally dissolved in 30 μL of nuclease-free water. RNA concentration and purity was measured using a Nanodrop (Denovix).

In order to enrich the bacterial RNA, the MICROBEnrich kit (ThermoFisher) was used. For each sample, 25 μg of RNA (spleen/bacteria mix) was processed according to the manufacturer's instructions. Of the resulting enriched bacterial RNA (free of eukaryotic RNA), 1 μg was reverse transcribed into cDNA (iScript Kit, Biorad) according to the manual. qPCRs were performed on a MyIQ cycler (Biorad). Briefly, 100 ng of cDNA, 500 nM of forward and reverse primer (see table below), respectively, were mixed with 10 μL of iTaq Universal SYBR Green Supermix (Biorad) and run for 50 cycles (denaturation 95 °C, 5 s; annealing/extension 55 °C, 30 s). The Ct values were normalized to the 16 S housekeeping gene (ΔCt) and plotted as fold change between infection conditions samples ($2^{-\Delta\Delta Ct}$). The primer pairs used were the following: EndoS: forward, AGAAGATACAGCAGGC GTAG, reverse, TCCCAACCACCTTTCTCTC; IdeS: forward, CTATCA-CACACCTACGCTAAC, reverse, CCAGCGGAATTAACACCAAC; 16 S: forward, ACCAAGGCGACGATACATAG, reverse, ACTCCCACCATCATT CTTCTC.

## IgG pulldowns

IgGs were purified in a 96 well plate setup using the Protein G Assay-MAP Bravo (Agilent) technology, according to the manufacturer's instructions. Briefly, 10 μl plasma was diluted with PBS to a final volume of 100 μl, and applied to pre-equilibrated Protein G columns. Columns were washed with PBS, and eluted in 0.1 M glycine (pH2). The final pH was neutralized with 1 M Tris, and saved until further use.

## Trypsin digestion and peptide desalting

Protein samples were resuspended in 8 M urea and reduced with 5 mM Tris(2-carboxyethyl)phosphine hydrochloride, pH 7.0 for 45 min at 37 °C, and alkylated with 25 mM iodoacetamide (Sigma) for 30 min at RT, followed by dilution with 100 mM ammonium bicarbonate to a final urea concentration below 1.5 M. Proteins were digested by incubation with trypsin (1/100, w/w, Sequencing Grade Modified Trypsin, Porcine; Promega) for at least 9 h at 37 °C. Digestion was stopped using 10% trifluoracetic acid (Sigma) to pH 2 to 3. Peptide clean-up was

performed by C18 reversed-phase spin columns according to manufacturer instructions (Silica C18 300 Å Columns; Harvard Apparatus). Solvents were removed using a vacuum concentrator (Genevac, miVac) and samples were resuspended in 50 µl HPLC-water (Fisher Chemical) with 2% acetonitrile and 0.2% formic acid (Sigma).

### LC-MS/MS for proteomics analysis
Peptide analyses (corresponding to 1 µg protein) were performed on a Q Exactive HF-X mass spectrometer (Thermo Fisher Scientific) connected to an EASY-nLC 1200 ultra-HPLC system (Thermo Fisher Scientific). Peptides were trapped on precolumn (PepMap100 C18 3 µm; 75 µm × 2 cm; Thermo Fisher Scientific) and separated on an EASY-Spray column (ES903, column temperature 45 °C; Thermo Fisher Scientific). Equilibrations of columns and sample loading were performed per manufacturer's guidelines. Mobile phases of solvent A (0.1% formic acid), and solvent B (0.1% formic acid, 80% acetonitrile) were used to run a linear gradient from 5% to 38% over 90 min at a flow rate of 350 nl/min. The variable window data independent acquisition (DIA) method is described by Bruderer et al.[48]. The data dependent acquisition (DDA) method was the manufacturer's default for 'high sample amount'. LC-MS performance was quality controlled with yeast protein extract digest (Promega). MS raw data was stored and managed by openBIS (v20.10.0) and converted to centroid indexed mzMLs with ThermoRawFileParser (v1.2.1).

### IgG glycoproteomics analysis
Purified IgG glycopeptides were analyzed on a Q Exactive HF-X mass spectrometer (Thermo Fisher Scientific) connected to an EASY-nLC 1200 ultra-HPLC system (Thermo Fisher Scientific). Peptides were trapped on precolumn (PepMap100 C18 3 µm; 75 µm × 2 cm; Thermo Fisher Scientific) and separated on an EASY-Spray column (Thermo Fisher Scientific). Mobile phases of solvent A (0.1% formic acid), and solvent B (0.1% formic acid, 80% acetonitrile) were used to run a linear gradient from 4 to 45% over 60 min. MS scans were acquired in data-dependent mode with the following settings, 60,000 resolution @ m/z 400, scan range m/z 600–1800, maximum injection time of 200 ms, stepped normalized collision energy (SNCE) of 15 and 35%, isolation window of 3.0 m/z, data-dependent HCD-MS/MS was performed for the ten most intense precursor ions.

### Bioinformatics and statistical analysis
**Proteomics data**. DIA MS files were processed using DIA-NN, version 1.8 and searched against the reference proteome (EMBL-EBI RELEASE 2019_04) EMBL mouse using DIA-NNs library-free mode using default parameters. Data was normalized using the cyclic loess normalization method in the NormalyzerDE tool[49]. All statistical methods were implemented in Python 3.6.10. The proteomics results were filtered using a one-way analysis of variance (ANOVA) followed by a Benjamini-Hochberg correction to control for a false discovery rate (FDR) of 0.10. Statistically significant identifications were subjected to principal-component analysis (PCA). Protein measurements were also standardized using a Z-score normalization. Proteomics results were analyzed separately using Welch's t test to generate volcano plots and heatmaps. Functional enrichment analysis of differentially abundant proteins was performed through Metascape[50]. Visualization of enriched Genome Ontology (GO) terms was produced using code adapted for TreeMap visualizations from the Web tool Revigo.

**Glycoproteomics data**. Raw files were searched in Byonic (Protein Metrics Inc, v5.0.3) integrated as a node in Proteome Discoverer (Thermo, v.2.5.0.400). Files were searched against a UniProt human and mouse protein database using the default search strategy: enzyme: trypsin with a maximum of two missed tryptic cleavages per peptide, up to one glycan per peptide as a 'rare' variable modification, up to 10/20 ppm deviation of the observed precursor/product ion masses from

the expected values, up to one Met oxidation ( + 15.994 Da) per peptide (variable 'common' modification), together with a predefined plasma glycan database of mouse and human structures and a decoy and contaminant database available in Byonic. Identifications with a Byonic score >100 were considered positive hits, and their spectra were manually validated. Both tryptic and semi-tryptic glycopeptide variants were collapsed into consensus glycopeptides and glycoforms were represented as a percent of total glycoform intensity for each glycosylation site. Heatmaps represent each glycosylation distribution present at a unique glycosylation site.

### Reporting summary
Further information on research design is available in the Nature Portfolio Reporting Summary linked to this article.

## Data availability
Raw mass spectrometry data associated with this manuscript is publicly available on MassIVE. Data can be found under the MassIVE **ID: MSV000092131**. Source data are provided with this paper.

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

## Acknowledgements

We thank Dr. Magnus Paulsson for assistance with qPCR analysis, and the Swedish Infrastructure for Biological Mass Spectrometry (BioMS) for assistance with MS analysis. This work was supported by funding from NIGMS (R35 GM119850 N.E.L., T32GM008806 J.T.S.) and the Novo Nordisk Foundation (NNF20SA0066621 N.E.L.). A.G.T., C.K., S.C. and J.M. were supported by the Wallenberg foundation (WAF grant number 2017.0271), the Swedish research council (grant number 2019-01646 and 2018-05795), the Viral and Bacterial Adhesin Network Training (ViBrANT) Program funded by the European Union's HORIZON 2020 Research and Innovation Program under the Marie Sklodowska-Curie Grant Agreement No 765042 and Alfred Österlunds Foundation.

## Author contributions

A.G.T. and J.M. conceived and conceptualized the project with significant input from of E.B. and O.S. A.G.T., E.B., E.V., S.C., B.O., J.D.E., M.C., and O.S. performed experiments and analysis. A.G.T., C.K., J.S., N.L., and J.M. analyzed and interpreted the mass spectrometry data. The

manuscript was mainly written by A.G.T. and J.M. with significant input from all co-authors.

## Funding

## Competing interests

The authors declare no competing interests.
