## [Peer Review File · Nature Communications]

Pathogen-driven degradation of endogenous and therapeutic antibodies during streptococcal infectionsREVIEWER COMMENTS

Reviewer #1 (Remarks to the Author):

Nature Communications manuscript NCOMMS-23-11560

This manuscript used sensitive proteomics and glycoproteomics techniques to identify the IgG deglycosylation and murine IgG3 cleavage by GAS virulence factors EndoS and IdeS. The experimental design and data interpretation of the Mass Spectrometry analysis and proteomic part, from a MS filed reviewer's point of view, is thorough and of high quality. I believe this is a very strong manuscript, I recommend publication in its current form.

Reviewer #2 (Remarks to the Author):

The manuscript entitled "Pathogen-driven degradation of endogenous and therapeutic antibodies in vivo during streptococcal infections" is a novel study which investigates the ability of the streptococcal protease IdeS and the endoglycosidase EndoS to alter human and mouse IgG both invitro and in vivo. The study importantly determines that these enzymes, in very elegant in vitro experiments, affect the structure and function of IgGs changing their ability to bind the FcR and removing certain specific carbohydrate residues as well as changing the size and function of the IgG. The affects of these two enzymes on the outcomes of in vivo in models of subcutaneous group A streptococcal infection as well as IP infection demonstrated different outcomes where IVIG was effective and not affected by these enzymes but in the subcutaneous infection the outcome of infection was greatly affected. The study of both natural IgG and IVIG in in vivo settings including these models of group A streptococcal infection are enlightening as to the importance of the enzymes on the infection outcome and in certain instances of therapeutic IVIG administration to alleviate diseases. Further the knowledge gained that may provide insights into vaccination and effective protection against group A streptococcal infections in humans and animal models. The authors should address the following comments:

1. The results are striking and the biochemical analyses in the overall study are meticulously performed and the data enlightening about the mechanisms and the overall reduction of IgG to a unit that is not as functional in vitro or in vivo the host. However, with that said, there are complicating factors in the host that should be discussed/addressed despite the controls for the experiments and the interesting data in the article. One concern is the overall effect that may not be accountable including affects of the IdeS on host proteases or the effects of host proteases on the two molecules based on their in vivo results. The effects of the host proteases and other environmental challenges to IdeS and EndoS on the outcomes of the experiments was not dealt with in the discussion and should be addressed. Host endogenous proteases in tissues or blood or other molecules that bind equally as well may alter the IgGs or the IdeS or the EndoS themselves promoting degradation. Albeit there were results that for the most part could be explained until the in vivo studies were performed and the IP model did not see the effects of the subcutaneous model. Proteolytic inhibitors may be effective in the blood to alter the streptococcal proteases as molecules like the trypsin inhibitors could be important in the overall scheme of things. What would happen in vivo if the IdeS and the EndoS were administered in the absence of infection? Is the activity of these enzymes inhibited by plasma Or serum? Or intracellular proteases? Are the invitro effects realistic of the group A streptococcal infections in the tissues. These are topics that need to be discussed at the very least.

2. IgG degradation was modified by the degree of infection. Could this possibly be affected or augmented by similar enzymes present in host cells or serum?

3. Does human or animal sera from group A streptococcal infections contain antibodies against either of these molecules which would then alter or affect the outcomes of infections?

4. Why was IgG3 primarily affected in the experiments invitro? Was this also In vivo in the skin

model? In the IgG depletion experiments, wouldn't the IgG bound to proteins also be depleted? This could have become misleading in some of the experiments assessing the proteins. It also could affect the outcome of disease whether or not the enzymes IdeS and EndoS are bound by other proteins/carbohydrates or lipids, affected by proteolytic enzyme inhibitors or glycosidase inhibitors which might come from the skin or serum or host cells in vivo and potentially in vitro if experiments use serum or other human tissue materials for the studies.

5. Could the IdeS and the EndoS activate host proteases and other enzymes or other proteins/molecules that would in turn affect the outcomes of their experiments or the disease itself?

6. In Fig 1, e, legend, the infected animals are the subcutaneous skin model or the IP model? List in the legend.

7. In the study on page 7, Line 215, reorganization of the plasma proteome and degradation of IgG should be clarified more for the reader and what this means for the overall scheme of things in the disease.

8. How can you know that the 2 streptococcal enzymes are not affected by host proteases or other host molecules? What about proteases from neutrophils or other host cell infiltrates? Could this affect the in vivo model?

9. Can these effects be neutralized by specific neutralizing Ab or by protease or glycosidase inhibitors in the invitro experiments? Do humans or animal models produce specific Abs against IdeS or EndoS? Are they neutralizing Abs? Would they be in the IVIG?

10. In the overall picture of the role of Ab against group A streptococcal infections, the protection that is most effective against GAS is type specific opsonic antibody which is mentioned in the discussion. However with that in mind, the IVIG utilized would not likely contain type specific Abs per se but it is also not known if one or more of the 1000+ individuals used to make the purified IgG used in commercial IVIG preparations might have such type specific opsonizing Ab. It would not be expected to protect totally against an infection in humans or animal models but it could have protective capacity up to a certain point as shown in the study particularly protection in the IP model. Perhaps this difference was due to factors/lipids in the skin that were not present in the blood that allowed for the IdeS and EndoS to function as observed in vitro the same as in skin but in the IP model there were factors that allowed clearance of the group A streptococci more readily than in the skin. Neutrophils being the most important clearance mechanism with type specific antibody. If the IVIG might have contained/ There should be some attempt to explain the model based on what is shown or further studies perhaps should be done to try to explain this outcome. Could one explanation be a lack of neutrophils in the subcutaneous model? Some attempt should be made to discuss more satisfactorily the differences that might have affected the two vivo models. Have others had similar findings in subcutaneous models vs systemic infection models? One of the main points here is was the clearance by the IVIG actually due to antistreptococcal Abs in the IVIG or was it due to something else? Was the IVIG opsonic or bactericidal to their streptococcal strain tested in the in vivo models?

11. Are there fundamental differences in mouse and human IgG that would affect the outcomes of disease based on their current data? This was mentioned in the results but no mention of what the fundamental differences actually are that might have impacted their study.

Reviewer #3 (Remarks to the Author):

The paper by Toledo et al adds to our understanding about S. pyogenes-induced-immunoglobulin de-glycosylation and digestion during infection. It is nicely written and well presented, The group have previously shown that, in vivo, EndoS can deglycosylate host immunoglobulin – this was shown in both human samples and in a progressively invasive infection model starting in

the skin (reference 16).

In this report they have used advanced techniques to more rigorously determine the nature of IgG deglycosylation that occurs in the same mouse model. Novel findings are (i) that EndoS can impact human IVIG (though IVIG cannot protect the mouse model) and (ii) that deglycosylation of IgG during *S. pyogenes* infection is specific to the invasive skin infection model and is not seen when *S. pyogenes* is administered intraperitoneally. The authors propose that the difference in glycans hydrolysis is most likely due to differential induction of EndoS (and/or IdeS) in the different infection settings.

The findings are intriguing and of course raise the question as to relevance in the clinical setting (in particular, what is the clinical equivalent of direct i.p. *S. pyogenes* injection?). Although pro-opsonic actions of IVIG were affected by EndoS, as the authors acknowledge, IVIG has virulence factor neutralizing actions which might be as, if not more, important for invasive *S. pyogenes*. This raises a question as to why does the IVIG not neutralize the streptococcal enzymes?

The main issues that the authors ideally should address are:

1. Specificity of findings (proof that EndoS and IdeS activity are responsible). While the observational data from the mouse samples during infection, and ex vivo assays are persuasive, use of isogenic mutants in the experiments shown in figures 1 and 2 would provide increased confidence.
2. It was not possible to detect *S. pyogenes* enzymes in plasma by proteomics, but the authors did not use bioassays of enzyme activity using samples from the mice (e.g. serum or tissue samples from the skin co-incubated with fresh serum from un-infected mice) to demonstrate that the activity produced during infection was sufficient to account for the findings of IgG proteolysis and deglycosylation in murine samples (figs 1 & 2). The in vitro assays undertaken demonstrate plausibility, however the team did use a bioassay approach when considering ex vivo digestion of human IVIG (fig 3), indicating that this might be possible.
3. Figure 3. The authors in these experiments do use an EndoS KO bacterium here to address whether EndoS is responsible for changes in IVIG (deglycosylation) when mixed with infected mouse plasma or when administered in actual infection. Reproducibility is key here; can the results for all mice be shown? I think panel (g) which shows quantification of different isoforms of glycosylated IgG1 and IgG2 (WT vs. KO) might help us, but this is not explicit from the legend (Quantification of human IgG1 and IgG2 Fc glycopeptides upon infection with wildtype vs EndoS (KO) GAS strains). How many mice; what do the error bars mean; and is this from mice that are infected and treated with IVIG, or is this an ex vivo mixing? Panel (h) suggests there might be four mice per group?
4. The difference between skin and i.p infection. The authors propose that there are differential *S. pyogenes* responses to the skin compared with the i.p route of infection that result in more or less EndoS being produced. Would it be straightforward to confirm this, if not already done? I can see proteomics detected EndoS and IdeS in the skin (suppl fig 4) But I could not see a comment about peritoneal fluid. Did the authors use RNAseq or targeted qRTPCR to confirm their hypothesis and then perhaps isolate samples from infected mice to determine if the bacterial response can be recapitulated in vitro.

As an observation, the AP1 skin infection model appears fairly severe and it is perhaps not surprising that neither EndoS KO or IVIG has any impact on the model. AP1 is an animal-passaged derivative of an M1 isolate that has a *covS* mutation, making it hypervirulent. The authors do not discuss whether systemic dissemination from the skin focus to systemic infection leads to additional mutations in vivo, that increase virulence, but this presumably may be possible. Many authors have proposed that host factors such as Mg²⁺, LL-37 (absent in mice however), and neutrophils trigger *covRS* mutations. It is likely however that a variety of innate pressures might trigger homeostatic adjustment of regulators like *covRS*.

As a minor point, I note that the EndoS mutant was made >20y ago and it might be prudent to confirm there are no other mutations and/or that there are no polar effects on adjacent genes.

REVIEWER COMMENTS

Reviewer #1 (Remarks to the Author):

Nature Communications manuscript NCOMMS-23-11560

This manuscript used sensitive proteomics and glycoproteomics techniques to identify the IgG deglycosylation and murine IgG3 cleavage by GAS virulence factors EndoS and IdeS. The experimental design and data interpretation of the Mass Spectrometry analysis and proteomic part, from a MS filed reviewer's point of view, is thorough and of high quality. I believe this is a very strong manuscript, I recommend publication in its current form.

Reviewer #2 (Remarks to the Author):

The manuscript entitled "Pathogen-driven degradation of endogenous and therapeutic antibodies in vivo during streptococcal infections" is a novel study which investigates the ability of the streptococcal protease IdeS and the endoglycosidase EndoS to alter human and mouse IgG both invitro and in vivo. The study importantly determines that these enzymes, in very elegant in vitro experiments, affect the structure and function of IgGs changing their ability to bind the FcR and removing certain specific carbohydrate residues as well as changing the size and function of the IgG. The affects of these two enzymes on the outcomes of in vivo in models of subcutaneous group A streptococcal infection as well as IP infection demonstrated different outcomes where IVIG was effective and not affected by these enzymes but in the subcutaneous infection the outcome of infection was greatly affected. The study of both natural IgG and IVIG in in vivo settings including these models of group A streptococcal infection are enlightening as to the importance of the enzymes on the infection outcome and in certain instances of therapeutic IVIG administration to alleviate diseases. Further the knowledge gained that may provide insights into vaccination and effective protection against group A streptococcal infections in humans and animal models. The authors should address the following comments:

1.The results are striking and the biochemical analyses in the overall study are meticulously performed and the data enlightening about the mechanisms and the overall reduction of IgG to a unit that is not as functional in vitro or in vivo the host. However, with that said, there are complicating factors in the host that should be discussed/addressed despite the controls for the experiments and the interesting data in the article. One concern is the overall effect that may not be accountable including affects of the IdeS on host proteases or the effects of host proteases on the two molecules based on their in vivo results. The effects of the host proteases and other environmental challenges to IdeS and EndoS on the outcomes of the experiments was not dealt with in the discussion and should be addressed. Host endogenous proteases in tissues or blood or other molecules that bind equally as well may alter the IgGs or the IdeS or the EndoS themselves promoting degradation. Albeit there were results that for the most part could be explained until the in vivo studies were performed and the IP model did not see the effects of the subcutaneous model. Proteolytic inhibitors may be effective in the blood to alter the streptococcal proteases as molecules like the trypsin inhibitors could be important in the overall scheme of things. What would happen in vivo if the IdeS and the EndoS were administered in the absence of infection? Is the activity of these enzymes inhibited by plasma, Or serum? Or intracellular proteases? Are the invitro effects realistic of the group A streptococcal infections in the tissues. These are topics that need to be discussed at the very least.

The reviewer raises a valid point. To firmly link the presence of EndoS to IgG deglycosylation, we repeated the infections using wildtype or isogenic EndoS mutant bacteria, and analyzed the level of deglycosylation of the endogenous IgG. That data has been added to Fig.1, showing that glycan truncation of murine IgG is completely abolished by knocking out EndoS. Additionally, we have also reinvestigated the plasma of S. aureus infected mice but we did not find any evidence of IdeS cleavage of murine

IgG3, pointing again towards the GAS specificity of this phenotype. In addition, to determine the differences in IdeS/EndoS activity observed between the two models, we extracted microbial mRNA from infected spleens and confirmed with qPCR that 1) IdeS and EndoS are both expressed in the subcutaneous model of infection, and 2) expression is significantly reduced in the intraperitoneal model of infection. This clearly correlates with the differential enzymatic activity of both IdeS and EndoS reported in this manuscript for each route of infection. This new data has been added to an updated Fig. 5, panel e.

It should also be pointed out that there are additional levels of evidence regarding the involvement of EndoS and IdeS in these phenotypes. Firstly, in a previous study (Reference 16) and using a completely different mass spectrometric technique (SRM), we already showed that deglycosylation in this mouse model is dependent on the presence of enzymatically active EndoS, as compared to isogenic mutants. We have added a new sentence to the results clarifying that, in reference to this previous study:

“The specificity of this EndoS-mediated phenotype was confirmed by control infection with isogenic mutant bacteria.”

Moreover, in this manuscript we also showed the impact of these enzymes on the cleavage of exogenous IVIG after infection with wt or mutant EndoS strain (Fig 3f-h). These results indicate that the cleavage of exogenous IVIG occurs in an EndoS-dependent manner. Also, the ex-vivo and in-vitro assays showed that the cleavage signatures of both EndoS and IdeS are the same as observed in vivo (Supplemental Fig.1, and Table.2). Finally, we showed that both enzymes are indeed expressed during infection at the protein level (Supplemental Fig.4)

With that being said, we also agree that we cannot completely rule out more indirect contributions of other endogenous factors in the modulation and regulation of these phenotypic switches. Therefore, we have now significantly expanded in the discussion to acknowledge that.

2. IgG degradation was modified by the degree of infection. Could this possibly be affected or augmented by similar enzymes present in host cells or serum?

- **Possibly but we have not found any evidence of that. On the other hand, we provide several levels of experimental data linking specifically IdeS and EndoS to IgG degradation, indicating that they are the major drivers of these phenotypic alterations (see above in point 1).**

3. Does human or animal sera from group A streptococcal infections contain antibodies against either of these molecules which would then alter or affect the outcomes of infections?

- **A very interesting issue raised by the reviewer. To address the comment, we used ELISA to show that there are no antibodies directed against these enzymes in murine plasma and added that data to a new supplemental fig 6. The rapid disease progression of both subcutaneous and intraperitoneal infections models that results in mortality by ~24-36h post infection is the most likely explanation for the absence of antibodies in these models. In ongoing work, we have observed that immunization through repeated exposure to low doses of immunogens is important to induce anti-GAS antibodies in the mouse. On the other hand, humans might indeed develop antibodies against GAS through natural exposure, but whether such antibodies can neutralize these enzymes is presently unclear. In fact, previous studies have shown that most people make IgG against IdeS and EndoS, but these antibodies do not interfere with the activity of the enzymes. To investigate this idea, we performed additional experiments to show the occurrence of relatively high antibody titers against EndoS and IdeS in IVIGs (supplemental fig 6). However, similarly to the previous studies, our data also shows that IVIGs are readily degraded by IdeS and EndoS, suggesting that these IgGs are**

incapable of blocking and neutralizing the enzymes. Possible reasons for this incapacity might be the occurrence of generally low antigen-specific IgG titers, suboptimal binding and/or reduced capacity of triggering downstream effector functions.

We have added a supplement figure and added a new paragraph to the discussion section where these results are discussed.

4. Why was IgG3 primarily affected in the experiments in vitro? Was this also in vivo in the skin model?

- In our work, we first noticed that IgG3 was more sensitive in the in vivo models (Fig. 2b and 2h). We then replicated that observation in vitro, using recombinant IdeS on human and murine plasma (Supplemental table. 2). We believe this specificity has to do with substrate differences between human and mouse IgGs (Fig. 2j), which is in line with previous studies.

We rephrased the following statement in the discussion and added a new reference:

“In fact, unlike human IgGs, most murine IgGs are known to be resistant to the proteolytic activity of IdeS, with the notable exception of IgG3, which we also found to be cleaved in the subcutaneous model of disseminating GAS infection, and subsequently cleared out from circulation. The basis for the species preferences of IdeS might lie in acquired amino acid variations across the hinge regions of human vs mouse IgG, but more studies are needed to clarify this, as well as the potential contribution of specific substrate residues to the catalytic efficiency of IdeS.”

In the IgG depletion experiments, wouldn't the IgG bound to proteins also be depleted? This could have become misleading in some of the experiments assessing the proteins.

- Depletion of proteins from plasma can lead to loss of other proteins as correctly pointed out by the reviewer. To mitigate this, we collected and analyzed both the IgG-enriched and the flow-through fractions to identify both endogenous and bacterial proteins in the respective fractions. As mentioned before, mouse IgGs do not bind streptococcal proteins since these animals are naïve and the rapid onset and sepsis development of the infection model does not allow for mounting an adaptive immune response. Small amounts of other endogenous proteins (i.e., complement) were still retained by Protein-G, but no truncated glycans were observed besides the IgG.

It also could affect the outcome of disease whether or not the enzymes IdeS and EndoS are bound by other proteins/carbohydrates or lipids, affected by proteolytic enzyme inhibitors or glycosidase inhibitors which might come from the skin or serum or host cells in vivo and potentially in vitro if experiments use serum or other human tissue materials for the studies.

- This is correct and would be a novel finding. To address this concern, we performed additional experiments due to the differential outcome of the IP model and the absence of glycan degradation. In these experiments, we spiked in recombinant EndoS into IP infected murine plasma, which resulted in pronounced IgG degradation under these conditions. The results demonstrate that there are no inhibitors circulating in IP plasma and that the absence of glycan degradation is simply due to the absence of enzyme in this model. This new data has now been added to the paper (supplemental fig. 7). Differential IdeS and EndoS expression in the SC vs IP model was also confirmed by qPCR analysis. That data has also been added to the revised manuscript (Fig. 5e).

Finally, we are not aware of any evidence of host factors affecting the activity of IdeS or EndoS, neither in our own studies or in the literature. In the subcutaneous model both enzymes are obviously active since we document degradation products. The same applies for the in vitro experiments using uninfected human and mouse plasma. This

evidence rules out that general plasma factors might interfere with enzyme activity, at least under these conditions.

5. Could the IdeS and the EndoS activate host proteases and other enzymes or other proteins/molecules that would in turn affect the outcomes of their experiments or the disease itself?

-See above (Point 1)

6. In Fig 1, e, legend, the infected animals are the subcutaneous skin model or the IP model? List in the legend.

-We have now fixed the figure legend to add that information

7. In the study on page 7, Line 215, reorganization of the plasma proteome and degradation of IgG should be clarified more for the reader and what this means for the overall scheme of things in the disease.

- We have now rephrased the statement in the following manner: “Collectively, our results demonstrate that disease progression in the GAS model is paralleled by profound alterations in the abundance of multiple plasma proteins, and a significant IdeS- and EndoS-dependent remodeling of IgG, resulting in proteolytic degradation of IgG3 and complete deglycosylation of the Fc region of all murine IgG subtypes.”

8. How can you know that the 2 streptococcal enzymes are not affected by host proteases or other host molecules? What about proteases from neutrophils or other host cell infiltrates? Could this affect the in vivo model?

- We have expanded the discussion to include that possibility, although most of the data points towards a main effect of IdeS and EndoS.

9. Can these effects be neutralized by specific neutralizing Ab or by protease or glycosidase inhibitors in the invitro experiments? Do humans or animal models produce specific Abs against IdeS or EndoS? Are they neutralizing Abs? Would they be in the IVIG?

- We have now looked for the presence of antibodies and added that data to the paper as stated above.

10. In the overall picture of the role of Ab against group A streptococcal infections, the protection that is most effective against GAS is type specific opsonic antibody which is mentioned in the discussion. However with that in mind, the IVIG utilized would not likely contain type specific Abs per se but it is also not known if one or more of the 1000+ individuals used to make the purified IgG used in commercial IVIG preparations might have such type specific opsonizing Ab. It would not be expected to protect totally against an infection in humans or animal models but it could have protective capacity up to a certain point as shown in the study particularly protection in the IP model. Perhaps this difference was due to factors/lipids in the skin that were not present in the blood that allowed for the IdeS and EndoS to function as observed in vitro the same as in skin but in the IP model there were factors that allowed clearance of the group A streptococci more readily than in the skin. Neutrophils being the most important clearance mechanism with type specific antibody. If the IVIG might have contained/ There should be some attempt to explain the model based on what is shown or further studies perhaps should be done to try to explain this outcome. Could one explanation be a lack of neutrophils in the subcutaneous model? Some attempt should be made to discuss more satisfactorily the differences that might have affected the two vivo models.

- See above (Point 1)

Have others had similar findings in subcutaneous models vs systemic infection models?

-Yes, differential proteome responses linked to routes of infection have been documented in GAS and other pathogens. We have added more references documenting this phenomenon and added the following statement to the discussion:

“More importantly, phenotypic differences linked to the route of infection and the host microenvironments have been documented both in GAS and other pathogens, indicating that context-sensitive regulation of the bacterial proteome in vivo might be a general phenomenon that contributes to the observed disease heterogeneity of bacterial infections. “

One of the main points here is was the clearance by the IVIG actually due to antistreptococcal Abs in the IVIG or was it due to something else?

- On a general level, the protective effect of IVIG is multifactorial and not only antibacterial but also immunomodulatory. For direct effects on bacterial pathogenesis, a combination of phagocytic killing but also neutralization of toxins contributes. For immunomodulatory effects occupation of Fc receptors, neutralization of cytokine storm, anti-apoptosis. The protection we see based on CFUs could be interpreted to reflect differences in the infection route between the two models. Clearance of bacteria from the blood may be more rapidly achieved in an IP model due to this route of administration that bypasses a local infection site to gain direct access to the blood. In the SC model a local infection develops where bacteria can have a more protected niche from which to continuously re-colonize the blood and disseminate to organs.

Was the IVIG opsonic or bactericidal to their streptococcal strain tested in the in vivo models?

- Yes, we have shown that in previous studies. We have now added a sentence to the results clarifying that and included the references as well.

11. Are there fundamental differences in mouse and human IgG that would affect the outcomes of disease based on their current data? This was mentioned in the results but no mention of what the fundamental differences actually are that might have impacted their study.

- We have rephrased the following paragraph in the discussion:

“In the same line, our current study highlights important species differences between human and mouse immune responses, including the expression of different IgG subclass distributions (i.e., IgG1-4 in humans vs IgG-1, -2b, -2c and -3 in mice) and Fc-glycosylation patterns (e.g., absence of NeuGc in humans), as well as the differential susceptibility of specific IgG subtypes to streptococcal virulence factors such as IdeS. In fact, unlike human IgGs, most murine IgGs are known to be resistant to the proteolytic activity of IdeS, with the notable exception of IgG3, which we also found to be cleaved in the subcutaneous model of disseminating GAS infection, and subsequently cleared out from circulation. The basis for the species preferences of IdeS might lie in acquired amino acid variations across the hinge regions of human vs mouse IgG, but more studies are needed to clarify this, as well as the potential contribution of specific substrate residues to the catalytic efficiency of IdeS.”

Reviewer #3 (Remarks to the Author):

The paper by Toledo et al adds to our understanding about S. pyogenes-induced-immunoglobulin de-glycosylation and digestion during infection. It is nicely written and well presented,

The group have previously shown that, in vivo, EndoS can deglycosylate host immunoglobulin – this was shown in both human samples and in a progressively invasive infection model starting in the skin (reference 16).

In this report they have used advanced techniques to more rigorously determine the nature of IgG deglycosylation that occurs in the same mouse model. Novel findings are (i) that EndoS can impact human IVIG (though IVIG cannot protect the mouse model) and (ii) that deglycosylation of IgG during *S. pyogenes* infection is specific to the invasive skin infection model and is not seen when *S. pyogenes* is administered intraperitoneally. The authors propose that the difference in glycans hydrolysis is most likely due to differential induction of EndoS (and/or IdeS) in the different infection settings.

The findings are intriguing and of course raise the question as to relevance in the clinical setting (in particular, what is the clinical equivalent of direct i.p. *S. pyogenes* injection?). Although pro-opsonic actions of IVIG were affected by EndoS, as the authors acknowledge, IVIG has virulence factor neutralizing actions which might be as, if not more, important for invasive *S. pyogenes*.

This raises a question as to why does the IVIG not neutralize the streptococcal enzymes?

The main issues that the authors ideally should address are:

1. Specificity of findings (proof that EndoS and IdeS activity are responsible). While the observational data from the mouse samples during infection, and ex vivo assays are persuasive, use of isogenic mutants in the experiments shown in figures 1 and 2 would provide increased confidence.

We agree with this comment. As similar concerns were raised by the other reviewers we have performed several additional experiments to demonstrate that EndoS and IdeS are the enzymes responsible for IgG protein and glycan degradation. First, we repeated the infections using wildtype vs isogenic EndoS mutant bacteria, and analyzed the level of deglycosylation of the endogenous IgG. That data has been added to Fig.1, showing that glycan truncation of murine IgG is completely abolished by knocking out EndoS. In addition, we extracted microbial mRNA from infected spleens and confirmed with qPCR that 1) IdeS and EndoS are both expressed in the subcutaneous model of infection, and 2) expression is significantly reduced in the IP model of infection. This clearly correlates with the differential enzymatic activity of both IdeS and EndoS reported in this manuscript for each route of infection. This new data is presented in Fig. 5e.

In Fig 3f-h, we also provided evidence for the cleavage of exogenous IVIG in an EndoS-dependent manner by contrasting with an isogenic EndoS mutant bacteria. Also, the ex-vivo and in-vitro assays showed that the cleavage signatures of both EndoS and IdeS are the same as observed in vivo (Supplemental Fig.1, and Table.2). We have also re-investigated the plasma of *S. aureus* infected mice but no signature of IdeS cleavage of murine IgG3 could be observed, pointing again towards the GAS specificity of this phenotype. We also show that both enzymes are indeed expressed during infection at the protein level (Supplemental Fig.4). Finally, in a previous study (Reference 16) and using a completely different mass spectrometric technique (SRM), we also showed that deglycosylation in this mouse model is dependent on the presence of enzymatically active EndoS, as compared to isogenic mutants. We have added a new sentence to the results clarifying that, in reference to this previous study:

“The specificity of this EndoS-mediated phenotype was confirmed by control infection with isogenic mutant bacteria.”

With that being said, we also agree that we cannot completely rule out more indirect contributions of other endogenous factors in the modulation and regulation of these phenotypic switches. Therefore, we have now significantly expanded in the discussion to acknowledge that.

2. It was not possible to detect *S. pyogenes* enzymes in plasma by proteomics, but the authors did not use bioassays of enzyme activity using samples from the mice (e.g. serum or tissue samples from the skin co-incubated with fresh serum from un-infected mice) to demonstrate that the activity produced during infection was sufficient to account for the findings of IgG proteolysis and deglycosylation in murine samples (figs 1 & 2). The in vitro assays undertaken demonstrate plausibility, however the team did use a bioassay approach when considering ex vivo digestion of human IVIG (fig 3), indicating that this might be possible.

Now we repeated the infections using wildtype vs isogenic EndoS mutant bacteria, and analyzed the level of deglycosylation of the endogenous IgG. That data has been added to Fig.1. In addition, we extracted microbial mRNA from infected spleens and confirmed with qPCR that IdeS and EndoS are both expressed in the subcutaneous model of infection, and their expression is significantly reduced in the IP model of infection. This correlates with the differential enzymatic activity of both IdeS and EndoS reported in this manuscript for each route of infection. This new data is presented in Fig. 5e.

3. Figure 3. The authors in these experiments do use an EndoS KO bacterium here to address whether EndoS is responsible for changes in IVIG (deglycosylation) when mixed with infected mouse plasma or when administered in actual infection. Reproducibility is key here; can the results for all mice be shown? I think panel (g) which shows quantification of different isoforms of glycosylated IgG1 and IgG2 (WT vs. KO) might help us, but this is not explicit from the legend (Quantification of human IgG1 and IgG2 Fc glycopeptides upon infection with wildtype vs EndoS (KO) GAS strains). How many mice; what do the error bars mean; and is this from mice that are infected and treated with IVIG, or is this an ex vivo mixing? Panel (h) suggests there might be four mice per group?

- We thank the reviewer for bringing this issue to our attention. Yes, the data in Fig. 3g correspond to animals infected and treated with IVIG, (n=4/condition). We have now clarified this information in the legends.

4. The difference between skin and i.p infection. The authors propose that there are differential *S. pyogenes* responses to the skin compared with the i.p route of infection that result in more or less EndoS being produced. Would it be straightforward to confirm this, if not already done? I can see proteomics detected EndoS and IdeS in the skin (suppl fig 4) But I could not see a comment about peritoneal fluid. Did the authors use RNAseq or targeted qRT-PCR to confirm their hypothesis and then perhaps isolate samples from infected mice to determine if the bacterial response can be recapitulated in vitro.

- We agree with the reviewer that this is important. We have now performed the experiment and confirmed the differential expression of IdeS and EndoS in the Sc vs IP model, explaining the absence of glycan and protein degradation in the IP model. This new data is presented in Fig. 5e.

As an observation, the AP1 skin infection model appears fairly severe and it is perhaps not surprising that neither EndoS KO or IVIG has any impact on the model. AP1 is an animal-passaged derivative of an M1 isolate that has a covS mutation, making it hypervirulent. The authors do not discuss whether systemic dissemination from the skin focus to systemic infection leads to additional mutations in vivo, that increase virulence, but this presumably may be possible. Many authors have proposed that host factors such as Mg²⁺, LL-37 (absent in mice however), and neutrophils trigger covRS mutations. It is likely however that a variety of innate pressures might trigger homeostatic adjustment of regulators like covRS.

- This is an intriguing possibility and might perhaps occur, although it should be more likely in the context of several passages of the bacteria and not on single point experiments as the ones in this study. However, we have added a sentence to the discussion:

“Although there is a possibility that new mutations might be acquired during our infections, this is most likely to occur over several passages of the same strain on the mice, which was not the case for our particular experimental design.”

As a minor point, I note that the EndoS mutant was made >20y ago and it might be prudent to confirm there are no other mutations and/or that there are no polar effects on adjacent genes.

- Based on sequencing and phenotypic characterization no other noticeable differences besides EndoS activity are present in this strain, as also shown in Fig.4a-f. But efforts are underway in the lab to produce more streptococcal *ndoS* mutants, with site directed mutation of the catalytic site with retained IgG-binding and deletions of the IgG-binding domain with an intact catalytic site. The activities of such recombinant forms of EndoS have already been elucidated and published, but not in the the context of streptococcal-host interactions.

REVIEWERS' COMMENTS

Reviewer #2 (Remarks to the Author):

The authors have made important revisions and added new data and experiments to the manuscript and study and have clearly addressed the reviewers comments. The paper is now quite acceptable and it is an important contribution to the area of streptococcal pathogenesis and also important in understanding how bacteria can affect antibodies and IVIG and the immune system. This article is a very important contribution.

The only comment is that the X and Y axes on the graph in supplemental figure 7 need to be described in the legend. It was a bit confusing.

Again, the authors have done a great job on the revisions and have addressed specifically the reviewers concerns.

Reviewer #3 (Remarks to the Author):

The authors have responded robustly to almost all the queries raised and clarified their findings. It would be helpful if the authors could comment if an assay for EndoS activity in infected mouse plasma would show EndoS activity if mixed with (exogenous) IgG; at present the authors still say that

"Multiple strategies to directly measure EndoS in infected mouse plasma, including targeted and untargeted mass spectrometry analysis, were unsuccessful, suggesting that the enzyme circulates at low levels. "

REVIEWERS' COMMENTS

Reviewer #2 (Remarks to the Author):

The authors have made important revisions and added new data and experiments to the manuscript and study and have clearly addressed the reviewers comments. The paper is now quite acceptable and it is an important contribution to the area of streptococcal pathogenesis and also important in understanding how bacteria can affect antibodies and IVIG and the immune system. This article is a very important contribution.

The only comment is that the X and Y axes on the graph in supplemental figure 7 need to be described in the legend. It was a bit confusing.

- Now we have added the requested information

Again, the authors have done a great job on the revisions and have addressed specifically the reviewers concerns.

Reviewer #3 (Remarks to the Author):

The authors have responded robustly to almost all the queries raised and clarified their findings. It would be helpful if the authors could comment if an assay for EndoS activity in infected mouse plasma would show EndoS activity if mixed with (exogenous) IgG; at present the authors still say that

"Multiple strategies to directly measure EndoS in infected mouse plasma, including targeted and untargeted mass spectrometry analysis, were unsuccessful, suggesting that the enzyme circulates at low levels. "

- Now we have added a comment to that section